# Categorical Distributional Reinforcement Learning
# with Kullback-Leibler Divergence: Convergence and Asymptotics

**Tyler Kastner** [1 2]  **Mark Rowland** [3]  **Yunhao Tang** [4 †]  **Murat A. Erdogdu** [1 2]  **Amir-massoud Farahmand** [5 6 1]

## Abstract

We study the problem of distributional reinforcement learning using categorical parametrisations and a KL divergence loss. Previous work analyzing categorical distributional RL has done so using a Cramér distance-based loss, simplifying the analysis but creating a theory-practice gap. We introduce a preconditioned version of the algorithm, and prove that it is guaranteed to converge. We further derive the asymptotic variance of the categorical estimates under different learning rate regimes, and compare to that of classical reinforcement learning. We finally empirically validate our theoretical results and perform an empirical investigation into the relative strengths of using KL losses, and derive a number of actionable insights for practitioners.

## 1. Introduction

The expected return is a core object in reinforcement learning, allowing for both the evaluation of an agent's behaviour, and providing a means of improving this behaviour (Sutton, 2018). The traditional approach to predicting the expected return is to directly model it as a mean-prediction regression problem. Distributional reinforcement learning algorithms take a different approach, instead predicting the full probability distribution of the random return; the expected return then emerges as a byproduct of this richer prediction. Agents making use of distributional reinforcement learning have enjoyed a variety of empirical successes (Bellemare et al., 2017; Hessel et al., 2018; Yang et al., 2019; Bodnar et al., 2020; Wurman et al., 2022).

In particular, *categorical* distributional reinforcement learn-

ing methods cast the return distribution prediction problem as one of classification. These methods have proven performant in a variety of large-scale settings.

Theoretical convergence analysis has also been established for categorical dynamic programming, and categorical temporal-difference learning using the Cramér loss (Rowland et al., 2018; Boeck & Heitzinger, 2022; Peng et al., 2024). However, most large-scale implementations of categorical temporal-difference learning use a KL loss, rather than Cramér loss. This is a crucial detail of large-scale implementations, but has not yet been theoretically analysed.

In this work, we study categorical temporal-difference learning with KL loss (KL-CTD) as a fundamental tabular algorithm for reinforcement learning in its own right. In Section 3, we present several empirical examples of intriguing behaviour of KL-CTD in comparison to classical TD learning, motivating our study. We then go on in Section 4 to establish a connection between KL-CTD and distributional dynamic programming algorithms, which allow us to characterise the long-term behaviour of KL-CTD conditional on its convergence. In Section 5, we study the question of convergence, and introduce a novel preconditioned variant of KL-CTD. We then provide theory on the finer-grained asymptotic fluctuations of value function estimates produced by KL-based categorical distributional algorithms in Section 6. Our results provide a variety of hypotheses about the behaviour of KL-CTD, when we should expect it to perform well relative to classical TD, and intuition on how to set hyperparameters such as learning rate. We then validate these hypotheses in a variety of experiments.

## 2. Background

We consider a Markov decision process (Puterman, 2014; Sutton, 2018) with finite state space $\mathcal{X}$, action space $\mathcal{A}$, transition matrix $P \in \mathbb{R}^{\mathcal{X} \times \mathcal{A} \times \mathcal{X}}$, reward kernel $\mathcal{R} : \mathcal{X} \times \mathcal{A} \to \mathscr{P}(\mathbb{R})$, and discount factor $\gamma$. We assume a fixed policy $\pi : \mathcal{X} \to \mathscr{P}(\mathcal{A})$, which gives rise to the policy transition matrix $P^\pi \in \mathbb{R}^{\mathcal{X} \times \mathcal{X}}$ (where $P^\pi(x'|x) = \sum_{a \in \mathcal{A}} \pi(a|x)P(x'|x,a)$) and the policy reward kernel $\mathcal{R}^\pi : \mathcal{X} \to \mathscr{P}(\mathbb{R})$ (where $\mathcal{R}^\pi(x) = \sum_{a \in \mathcal{A}} \pi(a|x)\mathcal{R}(x,a)$). The policy $\pi$ induces a distri-

---

[1]University of Toronto [2]Vector Institute [3]Google DeepMind [4]Meta Platforms, Inc.  [†]Work done while at Google Deepmind [5]Polytechnique Montréal [6]Mila. Correspondence to: Tyler Kastner <tkastner@cs.toronto.edu>, Mark Rowland <markrowland@google.com>.

*Proceedings of the 42nd International Conference on Machine Learning*, Vancouver, Canada. PMLR 267, 2025. Copyright 2025 by the author(s).

bution over trajectories $(X_t, A_t, R_t)_{t \geq 0}$, where for every $t \geq 0$ we have $A_t \sim \pi(\cdot \,|\, X_t)$, $R_t = R(X_t, A_t)$, and $X_{t+1} \sim P(\cdot \,|\, X_t, A_t)$. The agent's performance is summarized by the discounted return $\sum_{t \geq 0} \gamma^t R_t$. We use $G^\pi(x)$ to denote the random return when generating actions according to $\pi$ and beginning the trajectory in state $X_0 = x$, and write $\eta^\pi(x)$ for the distribution of this random variable. The expected return across trajectories is the value function

$$V^\pi(x) = \mathbb{E}\left[G^\pi(x)\right];$$

estimating this quantity is central to value-based RL.

## 2.1. Value estimation

The Monte Carlo (MC) approach to estimating $V^\pi$ performs online regression against sample returns from trajectories. More precisely, we maintain an estimate $V \in \mathbb{R}^{\mathcal{X}}$ of the value function $V^\pi$, and given a return $G \sim \eta^\pi(x)$ generated by following $\pi$ from an initial state $x$, we perform the update

$$V(x) \leftarrow V(x) + \alpha\left(G - V(x)\right), \tag{1}$$

where $\alpha > 0$ is a step size parameter.

Temporal-difference (TD) methods utilize bootstrapping in place of sample returns from completed trajectories. More precisely, having observed a sample transition $(x, R, X')$ generated by following the policy $\pi$, the full sample of the return $G$ in the MC update of Equation (1) is replaced with a *bootstrap* estimate $r + \gamma V(X')$, the intuition being that if $V \approx V^\pi$, then $\mathbb{E}[R + \gamma V(X')] \approx \mathbb{E}[G]$. This leads to the TD update equation

$$V(x) \leftarrow V(x) + \alpha\left(R + \gamma V(X') - V(x)\right). \tag{2}$$

Temporal-difference learning can deliver several benefits relative to MC estimation, such as reduced variance in regression targets, as well as the ability to learn from individual transitions rather than requiring full trajectories.

## 2.2. Categorical distributional reinforcement learning

As opposed to classical value-based RL algorithms that focus solely on estimating the expected return of a policy, distributional reinforcement learning algorithms learn a parametrized approximation of the return distributions, i.e., $\eta^\pi(x) = \mathrm{Dist}(G^\pi(x))$, for each $x \in \mathcal{X}$.

Various families of distributional RL algorithms have been proposed, making use of distinct parametric families of return distributions (Morimura et al., 2010b; Dabney et al., 2018). A popular class of methods are based on categorical parametrizations (Bellemare et al., 2017), in which the distributions $(\eta^\pi(x) : x \in \mathcal{X})$ are approximated through categorical distributions of the form

$$\eta(x) = \sum_{i=1}^m p_i(x)\delta_{z_i}, \tag{3}$$

where $z_1, \ldots, z_m \in \mathbb{R}$ are fixed locations, and $((p_i(x))_{i=1}^m : x \in \mathcal{X})$ are learnable probability parameters. Throughout this paper, we will consider the case where each vector $(p_i(x))_{i=1}^m$ is parametrized via a vector of logits $\phi(x) \in \mathbb{R}^m$, writing $p^\phi(x) = \mathrm{softmax}(\phi(x))$ (Bellemare et al., 2017).

**Categorical MC estimation.** The categorical counterpart to the Monte Carlo update in Equation (1) is based on the idea of regressing the distribution $\eta(x) = \sum_{i=1}^m p_i(x)\delta_{z_i}$ towards the empirically observed $\delta_G$. In practice, we typically have $G \notin \{z_1, \ldots, z_m\}$, and so the KL divergence between $\delta_G$ and $\eta(x)$ is infinite. To circumvent this issue, the outcome $G$ is first mapped to a *two-hot* distribution as illustrated in Figure 1 (Bellemare et al., 2017; Schrittwieser et al., 2020), producing the projected target distribution

$$\sum_{i=1}^m h_i(G)\delta_{z_i}. \tag{4}$$

The precise definition of the $(h_i)_{i=1}^m$ functions defining the probability masses in Equation (4) is given by

$$h_i(z) = \max\left(0, \min\left(\frac{z - z_{i-1}}{z_i - z_{i-1}}, \frac{z_{i+1} - z}{z_{i+1} - z_i}\right)\right)$$

for $2 \leq i \leq m - 1$, and special edge cases

$$h_1(z) = \min\left(1, \max\left(0, \frac{z_2 - z}{z_2 - z_1}\right)\right),$$

$$h_m(z) = \min\left(1, \max\left(0, \frac{z - z_{m-1}}{z_m - z_{m-1}}\right)\right),$$

which account for "clipping" outcomes that occur outside of the interval $[z_1, z_m]$ to the endpoints.

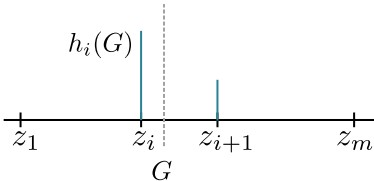

*Figure 1.* Projection of the outcome $G$ onto a distribution over the support set $\{z_1, \ldots, z_m\}$.

The *categorical Monte Carlo* (CMC) update, in analogy with the Monte Carlo update in Equation (1), can then be defined as

$$\phi(x) \leftarrow \phi(x) - \alpha \,\nabla_\phi \mathrm{KL}(h(G) \,||\, p^\phi(x)), \tag{5}$$

where we write $h(G) = (h_i(G))_{i=1}^m$ for the vector of two-hot probabilities corresponding to the outcome $G$.

**Categorical TD learning.** Just as in the case of value estimation described in Section 2.1, the CMC update can be modified to allow for learning from individual sample transitions $(x, R, X')$. As with classical TD learning, the

sampled return $G$ is replaced by a bootstrapped equivalent. Here, the random return $G$ is approximated by $R + \gamma Z'$, with $Z'$ sampled from the categorical approximation $\eta(X')$ to the return distribution at state $X'$. Setting up the KL gradient update as in Equation (5) for these bootstrapped samples, averaging according to the probability of sampling $Z' = z_i$ for $i = 1, \dots, m$, then yields the following update:

$$\phi(x) \leftarrow \phi(x) - \alpha \sum_{i=1}^{m} p_i^{\phi}(X') \times \tag{6}$$
$$\nabla_{\phi} \text{KL}(h(R + \gamma z_i) \, \| \, p^{\phi}(x)).$$

This is the categorical temporal-difference (CTD) learning update, proposed by Bellemare et al. (2017). This update forms a core component of many deep reinforcement learning agents utilizing distributional RL (Bellemare et al., 2017; Hessel et al., 2018). Our principal goal in this paper is to consider categorical temporal-difference learning as a fundamental, complementary approach to value prediction, and to understand its relative strengths and weaknesses as compared to classical TD learning.

**The Cramér-CTD update.** While the CTD update in Equation (6) has not previously been analysed, Rowland et al. (2018) prove convergence of an alternative version of categorical temporal-difference learning that performs a mixture update on probabilities directly, given by:

$$p(x) \leftarrow p(x) + \alpha \left( \sum_{i=1}^{m} p_i(X') h(R + \gamma z_i) - p(x) \right).$$

We refer to this variant as Cramér-CTD, due to the fact that it can be derived through the use of the Cramér distance (Cramér), rather than the KL, as a loss, and refer to the update in Equation (6) as KL-CTD, in contrast.

## 3. Motivation for the study of KL-CTD

Before beginning our analysis, we pause to motivate the study of KL-CTD, despite existing work studying Cramér-CTD (Rowland et al., 2018) and proving its equivalence to TD-learning in the tabular setting (Lyle et al., 2019). We will show unlike Cramér-CTD, KL-CTD has notable algorithmic differences to TD-learning, and understanding these differences is consequential as KL-CTD is the foundation of many performant deep RL algorithms (Bellemare et al., 2017; Hessel et al., 2018; Farebrother et al., 2024).

We present an example of learning curves for TD, Cramér-CTD, and KL-CTD on a particularly-chosen environment (see Appendix G) in Figure 2. Learning rates were tuned independently for each algorithm. The curves for TD and Cramér-CTD are perfectly overlapped, as predicted by Lyle et al. (2019). Further, KL-CTD exhibited significantly different learning dynamics, achieving a much better value MSE on this particular MDP.

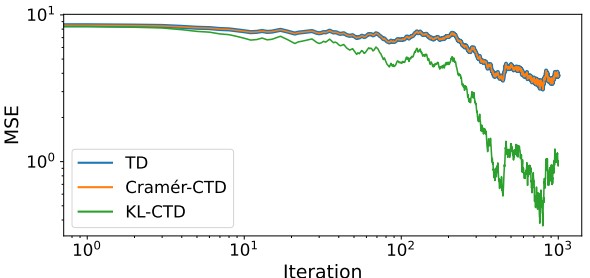

*Figure 2.* Mean-squared error of TD, Cramér-CTD, and KL-CTD with tuned learning rates.

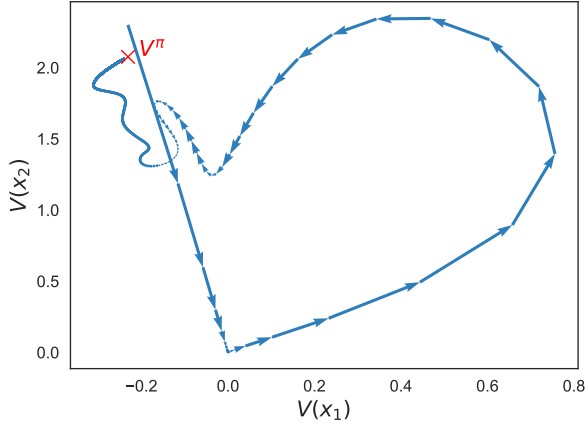

*Figure 3.* Value function dynamics of KL-CTD on a 2-state MDP.

Another striking difference of KL-CTD is the behaviour of its mean estimates. In particular, as a result of the loss being optimized is the KL divergence, which generally does not correspond to an improvement in the mean estimate. For probability measures $P, Q$ supported on $[a, b]$ with means $\mu_P, \mu_Q$ we always have that

$$|\mu_P - \mu_Q| \leq (b - a)\sqrt{2\text{KL}(P \, \| \, Q)},$$

so that optimizing the KL divergence will eventually optimize the mean estimate, however this is not guaranteed at each step. In the Monte Carlo setting, in expectation and under a suitable learning rate, the value error of the MC estimate is guaranteed to decrease at each step (see Appendix A). In contrast, we find the value error of the categorical Monte Carlo estimates may behave erratically as seen in Figure 3, entirely due to the dynamics optimizing the KL divergence.

## 4. A linear-algebraic perspective on KL-CTD

We begin by developing a deeper understanding of the KL-CTD update rule in Equation (6), and establishing connections with existing dynamic programming algorithms.

**Proposition 4.1.** *The expected KL-CTD update can be expressed via matrix-vector multiplication as*

$$\phi \leftarrow \phi + \alpha(T^\pi - I)p^\phi \,, \tag{7}$$

*where we interpret $\phi$ and $p^\phi$ as $|\mathcal{X}| \times m$-dimensional vectors, and where $T^\pi$ is the categorical distributional Bellman operator (Rowland et al., 2018; Bellemare et al., 2023; Rowland et al., 2024), given by*

$$T^\pi(x, i; \, y, j) = P^\pi(y \,|\, x) \, \mathbb{E}_{R \sim \mathcal{R}^\pi(x)}[h_i(R + \gamma z_j)] \,.$$

The connection that Proposition 4.1 establishes to the categorical distributional Bellman operator is important because a rich convergence theory exists for this operator (Rowland et al., 2018). Intuitively, repeated application of the operator $T^\pi$ to *any* initial collection of categorical probabilities $p \in \mathbb{R}^{\mathcal{X} \times m}$ results in convergence to a collection of probabilities $\tilde{p}^\pi$ such that the associated categorical distributions

$$\tilde{\eta}^\pi(x) = \sum_{i=1}^m \tilde{p}_i^\pi(x)\delta_{z_i}$$

form close approximations to the true return distributions $\eta^\pi(x)$. We recall the precise results required in the remainder of the paper below.

**Proposition 4.2** (Rowland et al., 2018). *The mapping $T^\pi : \Delta_m^{\mathcal{X}} \to \Delta_m^{\mathcal{X}}$ is a contraction with respect to the norm $\|\cdot\|$ defined by*

$$\|p\| = \max_{x \in \mathcal{X}} \|Cp(x)\|_2 \,, \tag{8}$$

*where $\|\cdot\|_2$ is the standard Euclidean norm, and $C$ is the lower-triangular matrix with $C_{ij} = 1$ if $i \geq j$, and $C_{ij} = 0$ otherwise. More precisely, for any $p, q \in \Delta_m^{\mathcal{X}}$, we have*

$$\|T^\pi p - T^\pi q\| \leq \sqrt{\gamma}\|p - q\| \,.$$

*As a result, $T^\pi$ has a unique fixed point $\tilde{p}^\pi \in \Delta_m^{\mathcal{X}}$. Writing $\tilde{\eta}^\pi \in \mathscr{P}(\mathbb{R})^{\mathcal{X}}$ for the corresponding distributions, so that*

$$\tilde{\eta}^\pi(x) = \sum_{i=1}^m \tilde{p}_i^\pi(x)\delta_{z_i} \,,$$

*we have that if $[z_1, z_m]$ contains all possible returns in the environment, $\tilde{\eta}^\pi(x)$ has the same mean as $\eta^\pi(x)$, and moreover,*

$$\ell_2^2(\tilde{\eta}^\pi(x), \eta^\pi(x)) \leq \frac{z_m - z_1}{(m-1)(1-\gamma)} \,,$$

*where $\ell_2$ is the Cramér distance (Cramér; Székely, 2003; Székely & Rizzo, 2013), defined by*

$$\ell_2(\nu, \nu') = \left( \int_{\mathbb{R}} (F_\nu(t) - F_{\nu'}(t))^2 \, \mathrm{d}t \right)^{1/2} \,,$$

*and $F_\nu$, $F_{\nu'}$ are the CDFs of $\nu, \nu'$, respectively.*

Interestingly, the form of Equation (7) shows that the (expected) KL-CTD updates are *linear in probability space*, and the nonlinearity of the updates is entirely due to the softmax transformation of the logits. We believe that this perspective may be interesting in its own right, as this form of nonlinearity is quite distinct from other nonlinear distributional algorithms, such as quantile-based. As a result of this connection between KL-CTD and dynamic programming, we conclude the following result.

**Proposition 4.3.** *The only stationary point of expected update in Equation (7) is the fixed point $\tilde{p}^\pi$ of $T^\pi$.*

Therefore, if KL-CTD converges to a stationary point, it must converge to $\tilde{p}^\pi$, and it thus produces accurate estimates of return distributions (with error as stated in Proposition 4.2), and if the support $\{z_1, \ldots, z_m\}$ is correctly selected to include all possible returns, then its mean predictions at convergence are exact. This is our first important finding: although the transient mean dynamics of KL-CTD may deviate significantly from those of classical TD (as shown in Section 3), at convergence it produces exact predictions, in contrast to other distributional approaches such as quantile TD (Dabney et al., 2018; Rowland et al., 2023).

## 5. Asymptotic convergence and preconditioned KL-CTD

We now turn our attention to questions of convergence for categorical algorithms using KL divergence.

### 5.1. Categorical Monte Carlo

The categorical MC update is straightforwardly analysable, as the following proposition shows.

**Proposition 5.1.** *If $G \sim \nu$, then the CMC update appearing in Equation (5) is a stochastic gradient for the objective $KL(\mathbb{E}_{G \sim \nu}[h(G)] \,||\, p^\phi)$, which is convex in $\phi$.*

Stochastic gradient descent on convex objectives is well-behaved theoretically (Bottou et al., 2018), and under mild conditions, it follows that iterative CMC updates with independent samples results in $p^\phi$ converging.

### 5.2. KL-CTD and preconditioning

Next, we consider the KL-CTD algorithm. In Section 2, we motivated this algorithm as a variant of CMC that introduces *bootstrapping*, a central idea in RL where the algorithm's own predictions are used as targets. Bootstrapping means that the algorithm updates can no longer be interpreted as stochastic gradient descent, and the question of whether such an algorithm converges is therefore more subtle.

**A heuristic argument for preconditioning.** A natural question is whether the expected updates of KL-CTD still act to reduce a quantity such as $\sum_{x \in \mathcal{X}} \mathrm{KL}(\tilde{p}^\pi(x) \,||\, p^\phi(x))$;

such an observation would form the basis of a proof of convergence for KL-CTD. However, this does not in fact hold in general; we give a numerical example in Appendix A (although we emphasize that this doesn't imply that KL-CTD does not converge). The issue stems from the fact that the expected change in this quantity is

$$\alpha \sum_{x \in \mathcal{X}} \langle \nabla_\phi \text{KL}(\tilde{p}^\pi(x) \,||\, p^\phi(x)), T^\pi p^\phi(x) - p^\phi(x) \rangle + O(\alpha^2)$$
$$= \alpha \langle p^\phi - \tilde{p}^\pi, T^\pi p^\phi - p^\phi \rangle + O(\alpha^2).$$

In general, the inner product appearing above may be positive, leading to an increase in the summed KLs. This is due to the fact that $T^\pi$ may be *strictly expansive* in the $\ell^2$ norm on return PMFs. However, the background theory in Proposition 4.2 shows that $T^\pi$ exhibits contractive behaviour in a *weighted* norm. This motivates *preconditioning* the KL-CTD update with a matrix that incorporates this weighting factor, so that expected updates under this new preconditioned rule reduce a KL-like measurement to $\tilde{p}^\pi$ by construction. We introduce this new algorithm below.

**Definition 5.2.** Given logits $\phi \in \mathbb{R}^{\mathcal{X} \times m}$ parametrising categorical approximations of return distributions as in Equation (3), and an observed transition $(x, R, X')$, the *preconditioned KL categorical temporal-difference learning* (PKL-CTD) update is defined by

$$\phi(x) \leftarrow \phi(x) - \alpha \sum_{i=1}^{m} p_i^\phi(X') \times \quad (9)$$
$$C^\top C \, \nabla_\phi \text{KL}(h(R + \gamma z_i) \,||\, p^\phi(x)).$$

We explore the empirical performance of this update in Section 7; in the remainder of this section, we now investigate its convergence properties, describing how to make the motivation intuition above precise. Our proof of convergence follows the Robbins-Siegmund theorem (Robbins & Siegmund, 1971) and a chaining argument (Thakoor et al., 2022); to understand the dynamics of the random updates in Equation (9), we first analyse the dynamics of the corresponding expected updates. By following Proposition 4.1, we obtain an expression for the expected PKL-CTD update in terms of the operator $T^\pi$:

$$\phi_{k+1}(x) = \phi_k(x) + \alpha \, C^\top C \left( (T^\pi - I) \, p^\phi \right)(x). \quad (10)$$

We then establish two results that are key in establishing the convergence of PKL-CTD. The first establishes a new weighted contraction property for $T^\pi$, mirroring analysis undertaken by Wu et al. (2023) in the case of 1-Wasserstein distance.

**Proposition 5.3.** *The mapping $T^\pi : \Delta_m^\mathcal{X} \to \Delta_m^\mathcal{X}$ is a contraction with respect to the norm $\|\cdot\|_\pi$: for any $p, q \in \Delta_m^\mathcal{X}$, we have*

$$\|T^\pi p - T^\pi q\|_\pi \leq \sqrt{\gamma} \|p - q\|_\pi,$$

*where $\|\cdot\|_\pi$ is the norm defined by $\|p\|_\pi = \sum_{x \in \mathcal{X}} d^\pi(x) \|Cp(x)\|_2$*

Next, we establish that a weighted sum of KLs between the CDP fixed-point $\tilde{p}^\pi$ and the approximation $p^\phi$ must strictly decrease when $p^\phi$ is updated with the continuous-time dynamics associated with Equation (10).

**Proposition 5.4.** *The continuous-time dynamics of Equation (10) converges, and a Lyapunov function for its convergence is*

$$L(\phi) = \sum_{x \in \mathcal{X}} d^\pi(x) \, \text{KL}(\tilde{p}^\pi(x) \,||\, p^\phi(x)), \quad (11)$$

*where $d^\pi$ is the stationary distribution of the policy $\pi$.*

With these results, we are now ready to state the central convergence theorem for PKL-CTD, in the case of synchronous updates. We discuss the generalization to the asynchronous updates setting in Appendix A.3.

**Theorem 5.5.** *Suppose that $(\phi_k)_{k \geq 0}$ is a sequence of logits generated according to synchronous PKL-CTD updates. That is, for each $k \geq 0$, we have independent transitions $(x, R_k^x, X_k^x)$ such that*

$$\phi_{k+1}(x) = \phi_k(x) + \alpha_k \times$$
$$C^\top C \left( \sum_{i=1}^{m} p_i^{\phi_k}(X_k^x) h(R_k^x + \gamma z_i) - p^{\phi_k}(x) \right).$$

*Further suppose that the stepsizes $(\alpha_k)_{k \geq 0}$ satisfy the Robbins-Munro conditions $\sum_{k=0}^{\infty} \alpha_k = \infty$ and $\sum_{k=0}^{\infty} \alpha_k^2 < \infty$. Then we have that $\phi_k$ converges in the sense that $p^{\phi_k}(x) \to \tilde{p}^\pi(x)$ for every $x \in \mathcal{X}$ almost surely.*

## 6. Analysis of value estimates

We now perform a theoretical investigation into the algorithms considered and introduced, and their efficacy as value estimates. We consider both Monte Carlo (Section 6.1) and TD (Section 6.2) settings, and these analyses yield theoretically-motivated insights for practitioners.

### 6.1. Monte Carlo setting

We begin by examining the Monte Carlo methods. For the results in this section, we will write $(V_k)_{k \geq 0}$ and $(\phi_k^{\text{CMC}})_{k \geq 0}$ for sequences of iterates generated by following Equation (1), Equation (5), respectively. Further, we will consider a Monte Carlo variant of PKL-CTD (preconditioned CMC; PCMC), defined by replacing the bootstrap samples in Equation (9) with a Monte Carlo sample $G$, yielding the update

$$\phi(x) \leftarrow \phi(x) - \alpha \, C^\top C \, \nabla_\phi \text{KL}(h(G) \,||\, p^\phi(x)). \quad (12)$$

We write $(\phi_k^{\mathrm{PCMC}})_{k\geq 0}$ for the sequence of iterates generated by following Equation (12). The stationary points for both CMC and PCMC are any logits such that the induced probabilities are equal to $\Pi_C \eta^\pi(x) = \mathbb{E}_{G\sim\eta^\pi(x)}[h(G)]$ at each state, which we will refer to as $\check{\eta}^\pi(x)$.

The categorical estimates are used to produce value estimates given by

$$V_{\phi_k}^{\mathrm{CMC}}(x) = z^\top p^{\phi_k^{\mathrm{CMC}}}(x), \; V_{\phi_k}^{\mathrm{PCMC}}(x) = z^\top p^{\phi_k^{\mathrm{PCMC}}}(x),$$

where $z = (z_i)_{i=1}^m \in \mathbb{R}^m$ is the vector of locations.

The following proposition establishes convergence properties for the decreasing step size case; the analysis builds on core results on asymptotic fluctuations in stochastic approximation theory (Borkar, 2008).

**Proposition 6.1.** *Suppose the iterates $(V_k)_{k\geq 0}$, $(\phi_k^{\mathrm{CMC}})_{k\geq 0}$, and $(\phi_k^{\mathrm{PCMC}})_{k\geq 0}$ were produced using step size $\alpha_k = \alpha_0 k^{-\beta}$ for $\beta \in (1/2, 1)$. Further suppose that $\check{p}_i^\pi(x) > 0$ for each $x, i \in \mathcal{X} \times [m]$. Then we have that*

$$k^{\beta/2}(V_k(x) - V^\pi(x)) \xrightarrow{d} \mathcal{N}(0, \sigma^2(x))$$

$$k^{\beta/2}(V_{\phi_k}^{\mathrm{CMC}}(x) - V^\pi(x)) \xrightarrow{d} \mathcal{N}\left(0, \tfrac{1}{2}z^\top J(\check{p}^\pi(x))^2 z\right)$$

$$k^{\beta/2}(V_{\phi_k}^{\mathrm{PCMC}}(x) - V^\pi(x)) \xrightarrow{d} \mathcal{N}\left(0, \tfrac{1}{2}u(x)^\top u(x)\right),$$

*where $J(p) = \mathrm{diag}(p) - pp^\top$, $u(x) = CJ(\check{p}^\pi(x))z$ and $\sigma^2$ is the variance of $\eta^\pi(x)$.*

The technical assumption on positivity of $\check{p}_i^\pi(x)$ avoids the case of divergence of logits to $\pm\infty$. For both CMC and PCMC, the form of the limiting variance may appear opaque. However, we can note that the variance of the projected return distribution $\check{p}^\pi(x)$ is given by

$$\sum p_i z_i^2 - \left(\sum p_i z_i\right)^2 = z^\top \left(\mathrm{diag}(\check{p}) - (\check{p}^\pi)(\check{p}^\pi)^\top\right) z$$
$$= z^\top J(\check{p}^\pi)z,$$

where we write $p_i = (\check{p}^\pi(x))_i$ for clarity. Hence the limiting variance of categorical Monte Carlo at a state $x$ differs from the variance of $\check{p}^\pi(x)$ by a multiplication of $J(\check{p}^\pi(x))$ in the quadratic form. We can now isolate two sources of difference between the limiting variance of CMC and MC: the difference due to the extra multiplication of $J(\check{p}^\pi(x))$, and the difference between the variance of the return distribution $\eta^\pi(x)$ and the variance of the projected return distribution $\check{p}^\pi(x)$. We begin by analyzing the former, and demonstrate that the extra factor of $J(\check{p}^\pi(x))$ produces a quantity which is strictly smaller than the variance, yet still continuous with respect to the variance.

**Proposition 6.2.** *Suppose $p = (p_i)_{i=1}^m$ is a collection of probabilities associated to the locations $z = (z_i)_{i=1}^m$, and let the variance of this categorical distribution be $\sigma^2$. Then there exists $\beta$ depending only on $p$ such that*

$$\beta\sigma^2 < z^\top J(p)^2 z < \sigma^2.$$

We next consider the second differentiator, that is the difference of the variances due to the Cramér projection.

**Proposition 6.3.** *Suppose $\nu$ is a probability measure whose support is contained in $[a, b]$, and $\mathrm{Var}_{Z\sim\nu}(Z) = \sigma^2$. Let $a = z_1, \ldots, z_m = b$ be $m$ equally spaced points, Then we have*

$$\mathrm{Var}_{Z\sim\Pi_C\nu}(Z) = \sigma^2 + E(\nu) \geq \sigma^2,$$

*where $0 \leq E(\nu) \leq \frac{(b-a)^2}{4(m-1)^2}$ is a quantity capturing the amount of projection in the map $\nu \mapsto \Pi_C\nu$.*

Proposition 6.3 indicates that the projection increases the variance incurred, but the amount of this increase is upper bounded by $\frac{(b-a)^2}{(m-1)^2}$. This suggests that both (i) choosing the interval $[a, b]$ as small as possible while still containing the all possible returns (so as to not bias the mean estimate), and (ii) increasing the number of locations will both lower the asymptotic variance of CMC.

**On the number of locations and learning rate.** We now consider the constant step size setting, and identify another phenomenon which has important implications for practitioners: the role of learning rate and number of locations are jointly related in the learning process, and should be jointly tuned. Concretely, we can write KL loss-based updates (both MC and TD) in the form

$$\phi \leftarrow \phi - \alpha \, D \, \nabla_\phi \mathrm{KL}\left(T \,\|\, p_\phi\right), \tag{13}$$

where $D$ is a general preconditioner and $T$ is a target distribution. The eventual mean estimate is then produced by $\langle p_\phi, z \rangle$. The map $\phi \mapsto p_\phi$ is $\frac{1}{\sqrt{m}}$-Lipschitz (see Proposition A.1), meaning that an update of order $\alpha$ on $\phi$ results in a change in the mean estimate of order $\frac{\alpha}{\sqrt{m}}$. Additionally, the gradient appearing in Equation (13) is $\frac{1}{\sqrt{m}}$-Lipschitz, so that as the number of locations increases, the loss landscape becomes proportionally flatter and larger step sizes can be afforded. Based on these points, we suggest scaling $\alpha$ by $\sqrt{m}$, as our analysis suggests that this allows for transferable learning dynamics across different values of $m$.

### 6.2. Temporal-difference setting

We next turn to temporal-difference learning algorithms. We will write $(V_k)_{k\geq 0}$ for a sequence of value estimates following Equation (2) and $(\phi_k^{\mathrm{PCTD}})_{k\geq 0}$ for a sequence of logits following Equation (9), with associated value estimates $V_{\phi_k}^{\mathrm{PCTD}}$. The following results allow us to compare exact asymptotic variances for TD and PKL-CTD; the result for TD is due to Wu et al. (2024), while the PKL-CTD result is our own.

**Proposition 6.4.** *(Wu et al., 2024) If the step sizes $(\alpha_k)_{k\geq 0}$ are given by $\alpha_k = \alpha_0 k^{-\beta}$, we have that*

$$k^{\beta/2}(V_k(x) - V^\pi(x)) \xrightarrow{d} \mathcal{N}(0, (A_\pi^{-1}\Sigma_{\mathrm{TD}}A_\pi^{-\top})_x),$$

*where we use the notation $A_\pi = I - \gamma P^\pi$ and $\Sigma_{TD}$ is the matrix $\mathbb{E}[zz^\top]$, $z = (R_x + \gamma V(X'_x) - V(x) : x \in \mathcal{X})$ where the expectation is over sample transitions $(x, R_x, X'_x)$.*

**Proposition 6.5.** *If the step sizes $(\alpha_k)_{k \geq 0}$ are given by $\alpha_k = \alpha_0 k^{-\beta}$, we have that*

$$k^{\beta/2}(V^{\mathrm{PCTD}}_{\phi_k}(x) - V^\pi(x)) \xrightarrow{d} \mathcal{N}\left(0, b_x^\top \Sigma_{\mathrm{PCTD}}(x)\, b_x\right),$$

*where $b_x = J(\tilde{p}^\pi(x))z$ and $\Sigma_{\mathrm{PCTD}}(x)$ is the unique solution $\Sigma$ of the Lyapunov equation*

$$\mathbf{A}(T^\pi - I)J(p)\Sigma + \Sigma J(p)((T^\pi)^\top - I)\mathbf{A} + \mathbf{A}J(p)\mathbf{A} = 0$$

*subject to $\Sigma_{mm} = 0$, where we write $p = \tilde{p}^\pi(x)$, $\mathbf{A} = \mathbf{C}^\top\mathbf{C}$, and $T^\pi$ as introduced in Proposition 4.1.*

This result allows us to compare the exact asymptotic variances of TD and PKL-CTD, however the forms of their variance are not as interpretable as the Monte Carlo setting. In particular, the limiting variance does not have a closed form, and must be solved for as the solution of a Lyapunov equation. There is no equivalent result for KL-CTD as we do not have a convergence guarantee, however, we can derive the asymptotic variance conditioned on convergence (see Proposition A.2).

# 7. Empirical evaluation

Having derived a number of theoretical insights into the comparisons between the considered algorithms, in this section we perform an empirical investigation to validate these findings in practice, and to better understand the relative strengths of TD, KL-CTD, and PKL-CTD in small-scale, controlled experiments. Full details for replication are provided in Appendix G.

## 7.1. Empirical demonstration of theoretical results

In this section, we empirically validate the theory developed in Section 6, as well as build intuition around the findings and their practical implications.

**Asymptotic Variance.** We begin by considering the asymptotic variance results of Proposition 6.1. We empirically validate the convergence in distribution of the normalized value errors $k^{\beta/2}(V^{\mathrm{CMC}}_{\phi_k}(x) - V^\pi(x))$ to its limit, and plot empirical histograms of these errors across 10,000 independent experiments in Figure 4. We find that after roughly $10^4$ iterations the theoretical density becomes a good fit for the empirical density.

**Size of the support.** Proposition 6.3 suggests that to minimize the increased variance due to the categorical projection, the support of the locations should be as small as possible while still containing all possible returns. To empirically study this effect, we consider an MDP with discount factor

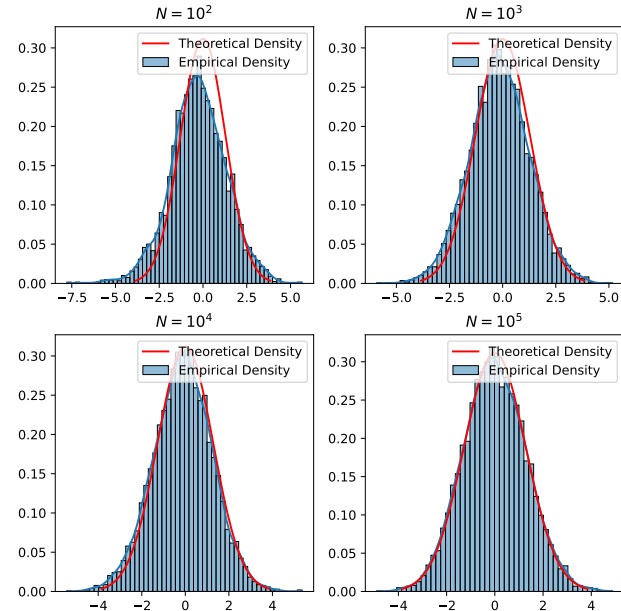

*Figure 4.* Comparison of theoretical and empirical normalized error distributions at different number of updates ($N$) for categorical Monte Carlo across 10,000 independent seeds.

$\gamma = 0.5$ and all reward distributions supported on $[0, 1]$, which ensures that all return distributions are supported on $[0, 2]$, and then sweep over possibilities for the final support location $z_m$, for a fixed number of support locations. We display the results from this procedure in Figure 5. The empirical results match our theoretical understanding: when $z_m < 2$, the approximate distribution cannot represent the full range of outcomes, resulting in biased mean estimates. Similarly, when $z_m > 2$, more variance is present in the Cramér projections, leading to value error.

**Scaling with number of atoms.** We next test our hypothesis

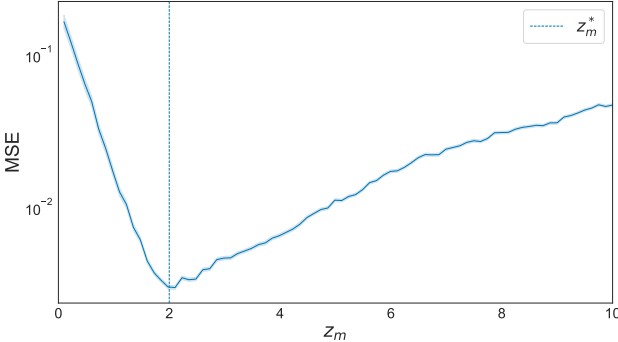

*Figure 5.* Value MSE against final support location for an environment where all return distributions are supported on $[0, 2]$. Each instance is run across 1,000 independent seeds.

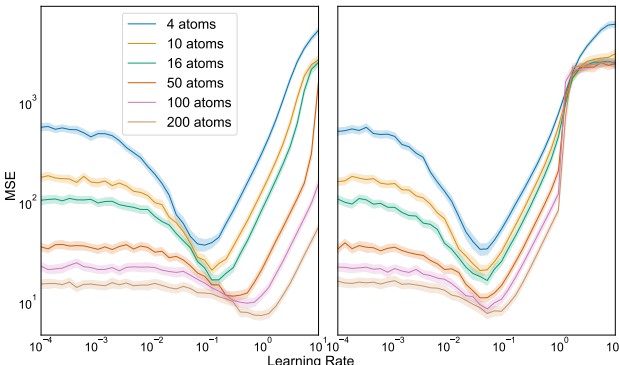

*Figure 6.* Value MSE against learning rate for CTD instances across varying number of locations. On the left no per-instance learning rate scaling is applied, on the right the learning rate is scaled by $\sqrt{m}$. Each instance is run across 200 independent seeds.

in Section 6.1 that the learning rate should be scaled by $\sqrt{m}$ in order to transfer across different numbers of atoms. To do this, we run CTD on a fixed MDP a number of times with different numbers of atoms, and for each number of atoms we sweep over 40 different learning rates and plot the MSE as a function of learning rate. We repeat this procedure once more, but for each number of atom locations $m$ we scale the learning rates used by $\sqrt{m}$. We present the result of these experiments in Figure 6. As was predicted by the theory, we can see that without applying learning rate scaling, larger number of support locations results in a larger optimal unscaled learning rate, however after scaling by $\sqrt{m}$, the optimal learning rate is similar across a wide range of values of $m$.

Another phenomenon demonstrated in Figure 6 is the monotonic decrease in MSE obtained as $m$ increases. This matches the behaviour predicted by the theory in Proposition 6.3, as increasing the number of atoms for a fixed support size minimizes the variance present in the Cramér projection.

### 7.2. Tabular experimental suite

In this section, we follow Rowland et al. (2023) and compare TD, KL-CTD, and PKL-CTD across a suite of MDPs with varying stochasticity in the transitions and rewards. We consider environments with deterministic transition structure ("Cycle"), sparse stochastic transitions ("Garnet"; Archibald et al., 1995), and dense stochastic transitions ("Dirichlet"), and deterministic/Gaussian/$t_2$-distributed rewards. For each MDP we perform a learning rate sweep for all methods over a fixed budget of 1,000 asynchronous updates with the environment, and calculate the MSE of the value estimates produced by the methods.

We present the results in Figure 7. In the deterministic do-

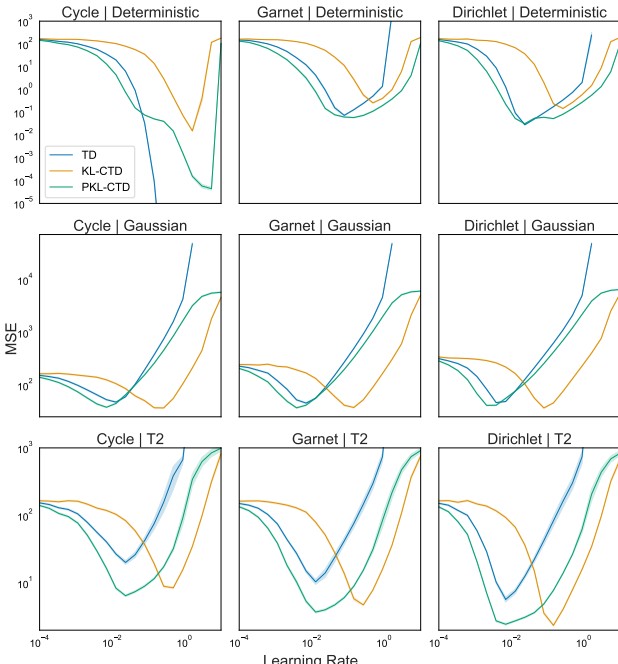

*Figure 7.* Comparison of tabular TD, KL-CTD, and PKL-CTD on environments with a variety of transition structures and rewards.

main (top-left), TD learning with a large learning rate is clearly the preferable approach. In other domains, however, there are benefits to using categorical approaches. In particular, in environments with high levels of reward noise, KL-CTD is preferable, and PKL-CTD performs well in all non-deterministic environments, indicating the benefits of an algorithm motivated by convergence considerations. We perform additional ablations over variations of the algorithms in Appendix F.

## 8. Conclusion

In this paper, we have studied the fundamental properties of categorical temporal-difference learning with KL-divergence. This has led us to propose a novel variant of categorical distributional RL, PKL-CTD, which makes use of preconditioning. We have proven convergence of PKL-CTD, and moreover analysed asymptotic variance of a variety of Monte Carlo and temporal-difference learning algorithms. These analyses have led to several practical insights, including the relationship between optimal learning rates and number of categories used in approximate distributions. Natural directions for further research include finite-sample analyses of KL-based categorical distributional algorithms.

## Impact Statement

This paper presents work whose goal is to advance fundamental reinforcement learning. There are many potential societal consequences of our work, none which we feel must be specifically highlighted here.

## Acknowledgements

The authors would like to thank the members of the Adage Lab, Jesse Farebrother, Harley Wiltzer, Kevin Li, and the anonymous reviewers for their helpful feedback and improvements to the paper. AMF acknowledges the support of NSERC through the Discovery Grant program [2021-03701]. MAE was partially supported by the NSERC Grant [2019-06167], the CIFAR AI Chairs program, and the CIFAR Catalyst grant. Resources used in preparing this research were provided, in part, by the Province of Ontario, the Government of Canada through CIFAR, and companies sponsoring the Vector Institute.

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

# APPENDICES

For convenience, we collect together the contents of the appendix:

- Appendix A contains several additional results referenced in the main text.
- Appendix B contains the proofs of results stated in Section 4.
- Appendix C contains the proofs of results stated in Section 5
- Appendix D contains the proofs of results stated in Section 6.
- Appendix E provides a summary of several areas of related work.
- Appendix F provides additional experimental results referenced in the main text.
- Appendix G contains full details for all experiments presented in the main paper.

## A. Additional Results

### A.1. Counterexample to KL as a Lyapunov function for KL-CTD

We construct a numerical example that shows that even in a single-state MDP, the KL divergence $\text{KL}(\tilde{p}^\pi(x) \,||\, p^\phi(x))$ need not decrease under KL-CTD dynamics. The example is constructed as follows.

We take a single-state, single-action MDP, such that taking the action in the state leads to a self-transition. We take the immediate reward to be $0.02644103$, and discount factor to be $0.99$. For the categorical support, we take 10 equally spaced atoms between 0 and 100. We consider initial values for the logit vector given by

$$(1.88191424, -0.02041108, -1.2244804, 0.44203928, 0.71425795,$$
$$0.46704711, 0.09271942, 0.11709652, -0.32497122, -1.59718562).$$

After taking a softmax, this yields initial categorical probabilities of approximately

$$(0.40581378, 0.06055603, 0.01816506, 0.09616057, 0.12624672,$$
$$0.09859566, 0.06780931, 0.06948262, 0.0446569, 0.01251333);$$

see Figure 8 for an illustration of these initial categorical probabilities $p_0$, the target distribution $T^\pi p_0$ under the categorical distributional Bellman operator, and the fixed-point categorical probabilities $\tilde{p}^\pi$.

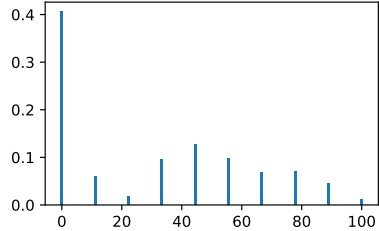 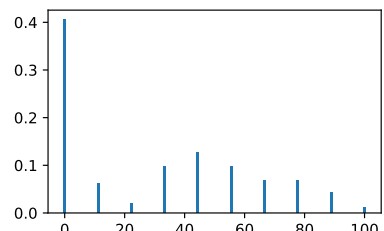 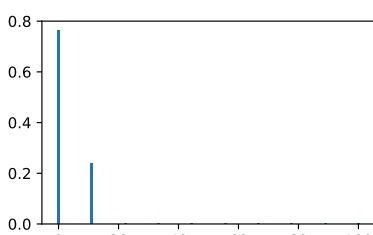

*Figure 8.* Left: The initial categorical probabilities $p_0$. Center: The initial target probabilities, $T^\pi p_0$. Right: The fixed-point categorical probabilities, $\tilde{p}^\pi$.

We simulate the cumulative effect of expected KL-CTD updates with small learning rate by numerically solving the flow

$$\partial_t \phi_t = (T^\pi - I)p^{\phi_t},$$

using the default `scipy.integrate.solve_ivp` method (Virtanen et al., 2020). We plot $\text{KL}(\tilde{p}^\pi \,||\, p^{\phi_t})$ against the ODE time $t$ in Figure 9, and note that initially, the KL divergence *increases*, meaning it is not a Lyapunov function for the KL-CTD dynamical system.

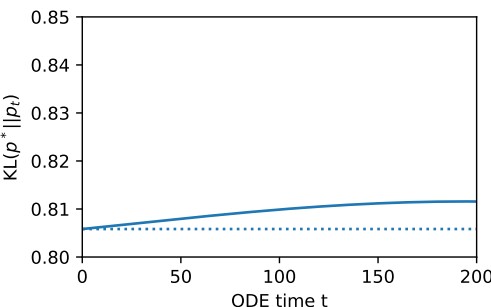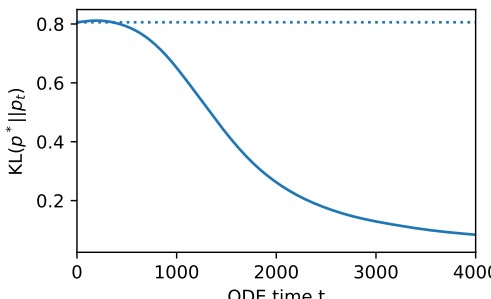

*Figure 9.* Numerical illustration of the non-monotonicity of the KL divergence for KL-CTD dynamics.

## A.2. Lipschitz coefficient of the KL gradient

**Proposition A.1.** *The gradient $\nabla_\varphi \mathrm{KL}(q \,||\, \mathrm{softmax}(\varphi))$ is Lipschitz with constant at most $\frac{\sqrt{m-1}}{m}$.*

*Proof.* We note that $\nabla_\varphi \mathrm{KL}(q \,||\, \mathrm{softmax}(\varphi)) = \mathrm{softmax}(\varphi) - q$ , and we also note the Jacobian of this is given by $J(\mathrm{softmax}(\varphi))$, where we use $J(p) = \mathrm{diag}(p) - pp^\top$. We next recall the fact that a function being Lipschitz with constant $L$ is equivalent to all eigenvalues of its Jacobian being less than $L$ at every point in its domain. We also recall the fact that the Frobenius norm of a matrix upper-bounds its largest eigenvalue. We now consider

$$\sup_{\varphi \in \mathbb{R}^m} \|J(p^\varphi)\|_F = \sup_{\varphi \in \mathbb{R}^m} \sqrt{\sum_{i=1}^m (p_i^\varphi)^2 (1 - p_i^\varphi)^2 + \sum_{i \neq j} (p_i^\varphi)^2 (p_j^\varphi)^2}$$
$$\leq \sqrt{\frac{m-1}{m^2}},$$

as the supremum is attained when $J(\mathrm{softmax}(\varphi))$ is a uniform distribution. Combining with the previous results, this implies that all eigenvalues of the Jacobian are upper bounded by $\frac{\sqrt{m-1}}{m}$, and we are complete. $\square$

**Proposition A.2.** *Suppose the iterates $(\phi_k^{\mathrm{CTD}})_{k \geq 0}$ are generated by following Equation (6) with step size $\alpha_k = \alpha_0 k^{-\beta}$ for $\beta \in (1/2, 1)$, and moreover suppose that this sequence converges. Let us write $V_k^{\mathrm{CTD}} = z^\top p^{\phi_k^{\mathrm{CTD}}}$. Then we have that*

$$k^{\beta/2}(V_k^{\mathrm{CTD}}(x) - V^\pi(x)) \xrightarrow{d} \mathcal{N}(0, z^\top J(\tilde{p}^\pi(x)) \Sigma_{\mathrm{CTD}}(x) J(\tilde{p}^\pi(x)) z),$$

*where $\Sigma_{\mathrm{CTD}}(x)$ is the unique solution of the Lyapunov equation*

$$(T^\pi - I) J(\tilde{p}^\pi(x)) \Sigma + \Sigma J(\tilde{p}^\pi(x)) ((T^\pi)^\top - I) + J(\tilde{p}^\pi) = 0$$

*subject to $\mathbf{1}^\top \Sigma \mathbf{1} = 0$.*

*Proof.* We first show that the logits must converge to a unique point. From Proposition 4.3 we know that we must have convergence to a $\phi_\star$ such that $p^{\phi_\star}(x) = \tilde{p}^\pi(x)$ for each $x$, which gives a 1-dimensional subspace of solutions for each $x$. However we also know that at each state $x$ the CTD update is orthogonal to $\mathbf{1}$, so that we must have $\mathbf{1}^\top(\phi_0^{\mathrm{CTD}}(x) - \phi_\star(x)) = 0$, which uniquely identifies $\phi_\star$. Since under our assumption we have a convergent sequence to a unique fixed point we satisfy the assumptions of Proposition D.1, which gives us that

$$k^{\beta/2}(\phi_k^{\mathrm{CTD}}(x) - \phi_\star(x)) \xrightarrow{d} \mathcal{N}(0, \Sigma_{\mathrm{TD}}(x)),$$

where $\Sigma_{\mathrm{TD}}(x))$ is a solution of the Lyapunov equation

$$(T^\pi - I) J(\tilde{p}^\pi(x)) \Sigma + \Sigma J(\tilde{p}^\pi(x)) ((T^\pi)^\top - I) + J(\tilde{p}^\pi) = 0.$$

This equation admits a 1-dimensional subspace of solutions, however we know that there is no covariance in the direction parallel to $\mathbf{1}$, so we can uniquely identify the solution through the constraint $\mathbf{1}^\top \Sigma \mathbf{1} = 0$. The remainder of the result now follows directly from applying Proposition D.2 to the map $\phi \mapsto z^\top p^\phi$, and we are complete. $\square$

## A.3. Extension to asynchronous PKL-CTD

Our convergence result in Section 5 focused on the case of asynchronous updates; here we discuss at a high-level how the key details map onto settings with asynchronous updates. In particular, from a sequence of transitions $(X_k, R_k, X'_k)_{k \geq 0}$ we compute a sequence of logits $(\phi_k)_{k \geq 0}$ given by

$$\phi_{k+1}(x) = \phi_k(x) + \alpha_{k,x} \, C^\top C \left( \sum_{i=1}^m p_i^{\phi_k}(X'_k) \, h(R_k + \gamma z_i) - p^{\phi_k}(x) \right)$$

for $x = X_k$, $\phi_{k+1}(x) = \phi_k(x)$ for $x \neq X_k$, and $\alpha_{k,x}$ is a state-dependent stepsize.

In this setting, under certain assumptions on the step sizes and distribution of updated states $(X_k)_{k \geq 0}$, the ODE becomes

$$\partial_t \phi_t(x) = c(x) \, C^\top C \, (T^\pi - I) p^{\phi_t}(x), \tag{14}$$

where $c(x)$ is a constant reflecting the relative update frequency of different states. It can be verified that these dynamics converge, using a Lyapunov function argument similar to that of Proposition 5.4.

**Proposition A.3.** *The ODE in Equation* (14) *converges, and a Lyapunov function for its convergence is*

$$L(\phi) = \sum_{x \in \mathcal{X}} \frac{d^\pi(x)}{c(x)} \mathrm{KL}(\tilde{p}^\pi(x) \| p^\phi(x))$$

*Proof.* Writing the right hand side of the ODE as $f(\phi)$, we can write out $\langle \nabla_\phi L(\phi), f(\phi) \rangle$:

$$
\begin{aligned}
\langle \nabla_\phi L(\phi), f(\phi) \rangle &= -\sum_{x \in \mathcal{X}} \frac{d^\pi(x)}{c(x)} \left\langle \tilde{p}^\pi(x) - p^\phi(x), c(x) \, C^\top C(T^\pi p^\phi(x) - p^\phi(x)) \right\rangle \\
&= -\sum_{x \in \mathcal{X}} d^\pi(x) \left\langle C(\tilde{p}^\pi(x) - p^\phi(x)), C(T^\pi p^\phi(x) - p^\phi(x)) \right\rangle,
\end{aligned}
$$

which becomes the same expression as in Appendix C.2, which is guaranteed to be negative. $\square$

A complete asynchronous convergence proof from here relies on the details on the sequence of transitions $(X_k, R_k, X'_k)$ (e.g. if $(X_k)_{k \geq 0}$ forms an ergodic Markov chain), and details on the step size sequence. Suitable results which can be used as a basis to build upon can be found in Borkar (1998) and Borkar & Meyn (2000).

# B. Proofs of results from Section 4

## B.1. Proof of Proposition 4.1

**Proposition 4.1.** *The expected KL-CTD update can be expressed via matrix-vector multiplication as*

$$\phi \leftarrow \phi + \alpha(T^\pi - I)p^\phi, \tag{7}$$

*where we interpret $\phi$ and $p^\phi$ as $|\mathcal{X}| \times m$-dimensional vectors, and where $T^\pi$ is the categorical distributional Bellman operator (Rowland et al., 2018; Bellemare et al., 2023; Rowland et al., 2024), given by*

$$T^\pi(x, i; \, y, j) = P^\pi(y \,|\, x) \, \mathbb{E}_{R \sim \mathcal{R}^\pi(x)}[h_i(R + \gamma z_j)].$$

*Proof.* To begin, recall that the synchronous sample-based KL-CTD update takes the form

$$\phi(x) \leftarrow \phi(x) - \alpha \sum_{i=1}^m p_i^\phi(X^x) \nabla_\phi \mathrm{KL}(h(R^x + \gamma z_i) \,\|\, p^\phi(x)),$$

where for each state $x \in \mathcal{X}$, $(x, R^x, X^x)$ is an independent transition sampled under $\pi$. Next, note that for a vector $\varphi \in \mathbb{R}^m$ and distribution $q \in \Delta_m$, we have

$$\nabla_\varphi \mathrm{KL}(q \,\|\, \mathrm{softmax}(\varphi)) = \mathrm{softmax}(\varphi) - q.$$

As a result, the expression for the update above can be rewritten as

$$\phi(x) \leftarrow \phi(x) + \alpha \sum_{i=1}^{m} p_i^{\phi}(X^x) \left( h(R^x + \gamma z_i) - p^{\phi}(x) \right).$$

Now, taking an expectation over the random transition $(x, R^x, X^x)$ yields

$$\mathbb{E}\left[ \sum_{j=1}^{m} p_i^{\phi}(X^x)(h(R^x + \gamma z_j) - p^{\phi}(x) \right]$$

$$= \sum_{j=1}^{m} \sum_{y \in \mathcal{X}} P^{\pi}(y|x) p_j^{\phi}(y) \mathbb{E}_{R \sim \mathcal{R}^{\pi}(x)}[h(R + \gamma z_j)] - p^{\phi}(x)$$

$$= \sum_{j=1}^{m} \sum_{y \in \mathcal{X}} P^{\pi}(y|x) \mathbb{E}_{R \sim \mathcal{R}^{\pi}(x)}[h(R + \gamma z_j)] p_j^{\phi}(y) - p^{\phi}(x)$$

$$= ((T^{\pi} - I)p^{\phi})(x),$$

as required. $\qquad\square$

### B.2. Proof of Proposition 4.3

**Proposition 4.3.** *The only stationary point of expected update in Equation* (7) *is the fixed point* $\tilde{p}^{\pi}$ *of* $T^{\pi}$.

*Proof.* At a stationary point, the expected update is 0. From Equation (7), the expected update (up to multiplication by the learning rate $\alpha$) takes the form $(T^{\pi} - I)p^{\phi}$, and thus the expected update being 0 implies $T^{\pi}p^{\phi} = p^{\phi}$. From the contraction theory recalled in Proposition 4.2, $T^{\pi}$ has a fixed point, $\tilde{p}^{\pi}$, and is also a contractive map on $\Delta_m^{\mathcal{X}}$, so does not have any other fixed points in $\Delta_m^{\mathcal{X}}$. Thus, at a stationary point, we must have $p^{\phi} = \tilde{p}^{\pi}$, as required. $\qquad\square$

## C. Proofs of results from Section 5

### C.1. Proof of Proposition 5.3

**Proposition 5.3.** *The mapping* $T^{\pi} : \Delta_m^{\mathcal{X}} \to \Delta_m^{\mathcal{X}}$ *is a contraction with respect to the norm* $\| \cdot \|_{\pi}$*: for any* $p, q \in \Delta_m^{\mathcal{X}}$*, we have*

$$\|T^{\pi}p - T^{\pi}q\|_{\pi} \leq \sqrt{\gamma}\|p - q\|_{\pi},$$

*where* $\| \cdot \|_{\pi}$ *is the norm defined by* $\|p\|_{\pi} = \sum_{x \in \mathcal{X}} d^{\pi}(x)\|Cp(x)\|_2$

*Proof.* Let $\eta, \mu \in \mathscr{P}(\mathbb{R})^{\mathcal{X}}$. We follow the structure of the proof of contractivity of $T^{\pi}$ in the norm $\| \cdot \|$ described in Proposition 4.2, and established by Rowland et al. (2018). Thus, we begin by observing that by identifying $\Delta_m$ with $\mathscr{P}(\{z_1, \ldots, z_m\})$ via the map $(p_i)_{i=1}^{m} \mapsto \sum_{i=1}^{m} p_i \delta_{z_i}$, the operator $T^{\pi}$ can be interpreted as the composition of the distributional Bellman operator $\mathcal{T}^{\pi} : \mathscr{P}(\{z_1, \ldots, z_m\})^{\mathcal{X}} \to \mathscr{P}(\mathbb{R})^{\mathcal{X}}$ and the categorical projection $\Pi_{\mathcal{C}} : \mathscr{P}(\mathbb{R}) \to \mathscr{P}(\mathbb{R})$ Rowland et al. (2018); see Bellemare et al. (2017); Morimura et al. (2010a); Chung & Sobel (1987) for earlier definitions of the distributional Bellman operator in alternative forms. We define the $\pi$-averaged version of the Cramér distance (see Proposition 4.2 as

$$\overline{\ell}_2^{\pi}(\eta, \eta') = \sum_{x \in \mathcal{X}} d^{\pi}(x)\ell_2(\eta(x), \eta'(x)),$$

for all $\eta, \eta' : \mathcal{X} \to \mathscr{P}(\mathbb{R})$. Then, arguing as in Rowland et al. (2018), we have

$$
\begin{aligned}
\overline{\ell}_2^\pi(\mathcal{T}^\pi\eta, \mathcal{T}^\pi\mu)^2 &= \sum_{x \in \mathcal{X}} d^\pi(x)\, \ell_2(\mathcal{T}^\pi\eta(x), \mathcal{T}^\pi\mu(x))^2 \\
&= \sum_{x \in \mathcal{X}} d^\pi(x)\, \ell_2\left(\int_{\mathbb{R}} \sum_{X' \in \mathcal{X}} (\mathrm{b}_{r,\gamma})_\# \eta(X')\, P^\pi(dr, X' \mid x), \int_{\mathbb{R}} \sum_{X' \in \mathcal{X}} (\mathrm{b}_{r,\gamma})_\# \mu(X')\, P^\pi(dr, X' \mid x)\right)^2 \\
&\leq \sum_{x \in \mathcal{X}} d^\pi(x) \int_{\mathbb{R}} \sum_{X' \in \mathcal{X}} P^\pi(dr, X' \mid x)\, \ell_2\left((\mathrm{b}_{r,\gamma})_\# \eta(X'), (\mathrm{b}_{r,\gamma})_\# \mu(X')\right)^2 \\
&= \gamma \sum_{x \in \mathcal{X}} d^\pi(x) \int_{\mathbb{R}} \sum_{X' \in \mathcal{X}} P^\pi(dr, X' \mid x)\, \ell_2\left(\eta(X'), \mu(X')\right)^2 \\
&= \gamma \sum_{X' \in \mathcal{X}} d^\pi(X')\, \ell_2\left(\eta(X'), \mu(X')\right)^2 \\
&\stackrel{(a)}{=} \gamma\, \overline{\ell}_2^\pi(\eta, \mu)^2 .
\end{aligned}
$$

Here, (a) follows from the invariance of $d^\pi$, as used in the argument of Wu et al. (2023) in the case of 1-Wasserstein distance. Further, $\Pi_C$ is a non-expansion in $\ell_2$ (Rowland et al., 2018), so together, we have that $\Pi_C \mathcal{T}^\pi$ is a $\sqrt{\gamma}$-contraction $\mathscr{P}(\{z_1, \ldots, z_m\})$ with respect to $\ell_2^\pi$ distance. Further, the map between $\Delta_m$ and $\mathscr{P}(\{z_1, \ldots, z_m\})$ is an isometry when $\Delta_m$ is equipped with the metric that sets the distance between $p, q \in \Delta_m$ to be $\|C(p - q)\|_2$, and $\mathscr{P}(\{z_1, \ldots, z_m\})$ is equipped with $\ell_2$. Hence, we recover that $T^\pi$ is a $\sqrt{\gamma}$-contraction on $\Delta_m^{\mathcal{X}}$ with respect to $\|\cdot\|_\pi$, as required. $\qquad\square$

## C.2. Proof of Proposition 5.4

**Proposition 5.4.** *The continuous-time dynamics of Equation* (10) *converges, and a Lyapunov function for its convergence is*

$$
L(\phi) = \sum_{x \in \mathcal{X}} d^\pi(x)\, \mathrm{KL}\big(\tilde{p}^\pi(x) \,\|\, p^\phi(x)\big), \tag{11}
$$

*where $d^\pi$ is the stationary distribution of the policy $\pi$.*

*Proof.* We can note that the gradient $\nabla_\phi L(\phi)$ is given by

$$
\nabla_\phi L(\phi) = \sum_{x \in \mathcal{X}} d^\pi(x)(p^\phi(x) - \tilde{p}^\pi(x)).
$$

Writing the right hand side of Equation (10) as $f(\phi)$, we can now write out $\langle \nabla_\phi L(\phi), f(\phi) \rangle$:

$$
\begin{aligned}
\langle \nabla_\phi L(\phi), f(\phi) \rangle &= -\sum_{x \in \mathcal{X}} d^\pi(x) \left\langle \tilde{p}^\pi(x) - p^\phi(x), C^\top C(T^\pi p^\phi - p^\phi(x)) \right\rangle \\
&= -\sum_{x \in \mathcal{X}} d^\pi(x) \left\langle C(\tilde{p}^\pi(x) - p^\phi(x)), C(T^\pi p^\phi - p^\phi(x)) \right\rangle \\
&= -\sum_{x \in \mathcal{X}} d^\pi(x) \left( \|C(\tilde{p}^\pi(x) - p^\phi(x))\|_2^2 - \left\langle C(\tilde{p}^\pi(x) - p^\phi(x)), C(\tilde{p}^\pi(x) - T^\pi p^\phi(x)) \right\rangle \right). \tag{$\star$}
\end{aligned}
$$

We can next note that

$$\sum_{x \in \mathcal{X}} d^\pi(x) \left\langle C(\tilde{p}^\pi(x) - p^\phi(x)), C(\tilde{p}^\pi(x) - T^\pi p^\phi(x)) \right\rangle$$

$$\stackrel{(a)}{\leq} \sum_{x \in \mathcal{X}} d^\pi(x) \left\| C(\tilde{p}^\pi(x) - p^\phi(x)) \right\|_2 \left\| C(\tilde{p}^\pi(x) - T^\pi p^\phi(x)) \right\|_2$$

$$= \sum_{x \in \mathcal{X}} \left( d^\pi(x)^{\frac{1}{2}} \left\| C(\tilde{p}^\pi(x) - p^\phi(x)) \right\|_2 \right) \left( d^\pi(x)^{\frac{1}{2}} \left\| C(\tilde{p}^\pi(x) - T^\pi p^\phi(x)) \right\|_2 \right)$$

$$\stackrel{(a)}{\leq} \left( \sum_{x \in \mathcal{X}} d^\pi(x) \left\| C(\tilde{p}^\pi(x) - p^\phi(x)) \right\|_2^2 \right)^{\frac{1}{2}} \left( \sum_{x \in \mathcal{X}} d^\pi(x) \left\| C(\tilde{p}^\pi(x) - T^\pi p^\phi(x)) \right\|_2^2 \right)^{\frac{1}{2}}$$

$$\stackrel{(b)}{\leq} \sqrt{\gamma} \left( \sum_{x \in \mathcal{X}} d^\pi(x) \left\| C(\tilde{p}^\pi(x) - p^\phi(x)) \right\|_2^2 \right)^{\frac{1}{2}} \left( \sum_{x \in \mathcal{X}} d^\pi(x) \left\| C(\tilde{p}^\pi(x) - p^\phi(x)) \right\|_2^2 \right)^{\frac{1}{2}}$$

$$= \sqrt{\gamma} \sum_{x \in \mathcal{X}} d^\pi(x) \left\| C(\tilde{p}^\pi(x) - p^\phi(x)) \right\|_2^2 ,$$

where both inequalities marked (a) follow from the Cauchy-Schwarz inequality, and (b) follows from the contractivity established in Proposition 5.3. This shows that the right hand side of $(\star)$ is strictly negative when $\hat{p}^\pi \neq p^\phi$, and hence we are complete. $\qquad\square$

## C.3. Proof of Theorem 5.5

**Theorem 5.5.** *Suppose that* $(\phi_k)_{k \geq 0}$ *is a sequence of logits generated according to synchronous PKL-CTD updates. That is, for each* $k \geq 0$*, we have independent transitions* $(x, R_k^x, X_k^x)$ *such that*

$$\phi_{k+1}(x) = \phi_k(x) + \alpha_k \times$$
$$C^\top C \left( \sum_{i=1}^m p_i^{\phi_k}(X_k^x) h(R_k^x + \gamma z_i) - p^{\phi_k}(x) \right).$$

*Further suppose that the stepsizes* $(\alpha_k)_{k \geq 0}$ *satisfy the Robbins-Munro conditions* $\sum_{k=0}^\infty \alpha_k = \infty$ *and* $\sum_{k=0}^\infty \alpha_k^2 < \infty$*. Then we have that* $\phi_k$ *converges in the sense that* $p^{\phi_k}(x) \to \tilde{p}^\pi(x)$ *for every* $x \in \mathcal{X}$ *almost surely.*

*Proof.* The high-level structure of the proof follows the classical Robbins-Siegmund theorem (Robbins & Siegmund, 1971), using the fact that as established in Proposition 5.4, the function $L$ in Equation (11) is a Lyapunov function for the associated continuous-time dynamical system, and combining this with the supermartingale convergence theorem. We begin with the assumption that $d^\pi$, a stationary distribution of $\pi$, has full support over the state space $\mathcal{X}$. We will then treat the more general case where this assumption does not hold by following the argument made in Thakoor et al. (2022), by making an inductive chaining argument across the communicating classes of the Markov chain induced on $\mathcal{X}$ by $\pi$.

To begin, we perform a second-order Taylor expansion of the Lyapunov function $L(\phi_{k+1})$ around $\phi_k$, using $\mathbb{E}_k$ to denote conditional expectation given the random variables defining the sequence $(\phi_l)_{0 \leq l \leq k}$, and $p_k$ to denote $p^{\phi_k}$:

$$\mathbb{E}_k \big[ L(\phi_{k+1}) \big] = \mathbb{E}_k \big[ L(\phi_k + \alpha_k \mathbf{C}^\top \mathbf{C}(T^\pi - I) p_k + \alpha_k \varepsilon_k) \big]$$
$$= L(\phi_k) + \alpha_k \langle \nabla L(\phi_k), \mathbf{C}^\top \mathbf{C}(T - I) p_k \rangle$$
$$\qquad + \alpha_k^2 \mathbb{E}_k \big[ \langle \mathbf{C}^\top \mathbf{C}(T^\pi - I) p_k + \varepsilon_k, HL(\phi_k) \mathbf{C}^\top \mathbf{C}(T^\pi - I) p_k + \alpha_k \varepsilon_k \rangle \big]$$
$$\leq L(\phi_k) - \alpha_k (1 - \sqrt{\gamma}) \| \tilde{p}^\pi - p_k \|_\pi^2$$
$$\qquad + \alpha_k^2 \mathbb{E}_k \big[ \langle \mathbf{C}^\top \mathbf{C}(T^\pi - I) p_k + \varepsilon_k, HL(\phi_k) \mathbf{C}^\top \mathbf{C}(T^\pi - I) p_k + \alpha_k \varepsilon_k \rangle \big] ,$$

where $HL(\phi_k)$ is the Hessian of $L$ at $\phi_k$, $\varepsilon_k(x) = C^\top C(T^\pi - I) p_k(x) - C^\top C \left( \sum_{i=1}^m p_i^{\phi_k}(X_k^x) h(R_k^x + \gamma z_i) - p_i^{\phi_k}(x) \right)$ is the zero-mean noise at step $k$, and the inequality follows from the argument in the proof of Proposition 5.4. The Hessian

of the Lyapunov function is uniformly bounded, and so too are the quantities $\mathbf{C}^\top \mathbf{C}(T^\pi - I)p_k + \varepsilon_k$, so there exists a global constant $B > 0$ such that

$$\mathbb{E}_k\big[L(\phi_{k+1})\big] \le L(\phi_k) - \alpha_k(1 - \sqrt{\gamma})\|\tilde{p}^\pi - p_k\|_\pi^2 + \alpha_k^2 B\,.$$

This is almost a supermartingale inequality, save for the final term on the right-hand side. The idea, however, is that thanks to the summability condition $\sum_{k\ge 0}\alpha_k^2 < \infty$, the cumulative effect of these terms does not interfere with the convergence guarantees associated with non-negative supermartingales; the following argument follows the approach of Robbins & Siegmund (1971).

Writing $L_k = L(\phi_k)$, note that if we define

$$\tilde{L}_k = L_k - \sum_{l=0}^{k-1}\alpha_l^2 B + \sum_{l=0}^{k-1}\alpha_l(1 - \sqrt{\gamma})\|\mathbf{C}(\tilde{p}^\pi - p_l)\|_{d^\pi}^2\,,$$

then we have

$$\mathbb{E}_k[\tilde{L}_{k+1}] = \mathbb{E}_k[L_{k+1}] - \sum_{l=0}^{k}\alpha_l^2 p_k + \sum_{l=0}^{k}\alpha_l(1 - \sqrt{\gamma})\|\tilde{p}^\pi - p_k\|_\pi^2$$

$$\le L_k - \alpha_k(1 - \sqrt{\gamma})\|\tilde{p}^\pi - p_k\|_\pi^2 + \alpha_k^2 p_k - \sum_{l=0}^{k}\alpha_l^2 p_k + \sum_{l=0}^{k}\alpha_l(1 - \sqrt{\gamma})\|\tilde{p}^\pi - p_l\|_\pi^2$$

$$= L_k - \sum_{l=0}^{k-1}\alpha_l^2 p_k + \sum_{l=0}^{k-1}\alpha_l(1 - \sqrt{\gamma})\|\tilde{p}^\pi - p_l\|_\pi^2$$

$$= \tilde{L}_k\,.$$

Hence, $(\tilde{L}_k)_{k\ge 0}$ is a supermartingale, and it is uniformly bounded below by $-p_k \sum_{k=0}^{\infty}\alpha_k^2$, and so by the martingale convergence theorem, we deduce that $\tilde{L}_k$ converges almost surely. It therefore follows that

$$\sum_{l=0}^{k}\alpha_l(1 - \sqrt{\gamma})\|\tilde{p}^\pi - p_l\|_\pi^2$$

converges almost surely, and hence so does $L_k$. If $L_k$ does not converge to 0 almost surely, then there exists $\varepsilon > 0$ such that $\liminf_k L_k > \varepsilon$ with positive probability. On this event, we would therefore have that

$$\sum_{l=0}^{k}\alpha_l(1 - \sqrt{\gamma})\|\tilde{p}^\pi - p_l\|_\pi^2$$

diverges, since $\sum_{l=0}^{\infty}\alpha_l = \infty$, a contradiction. Hence, $L_k \to 0$ almost surely, and this implies $p_k \to \tilde{p}^\pi$ almost surely, as required, since we assume $d^\pi$ has full support.

To make the general argument, where $d^\pi$ does not have full support, we can proceed in exactly the same manner as Thakoor et al. (2022). First, by adjoining a terminal state if necessary (to deal with environments in which episodes terminate in finite time), there exists an invariant probability distribution $d^\pi$ over the state space $\mathcal{X}$ under the transition dynamics induced by $\pi$. The argument above applies verbatim to obtain $\|\tilde{p} - p^{\phi_k}\|_\pi \to 0$ (though as $d^\pi$ does not have full support, this does not show $p_k \to \tilde{p}^\pi$).

Now, we apply an inductive argument to the communicating classes of the Markov chain on $\mathcal{X}$ induced by $\pi$. The set of communicating classes of a finite state Markov chain form a directed acyclic graph (with a directed edge drawn from one communicating class $K_1$ to another $K_2$ if there exist $x \in K_1, y \in K_2$ such that $P^\pi(y|x) > 0$. The leaves of this directed acyclic graph are precisely the communicating states which have an invariant probability distribution. For any such invariant class $K$, there exists an invariant distribution $d^\pi$ supported on $K$, and the argument above applies to obtain $p_k(x) \to \tilde{p}^\pi(x)$ for all $x \in K$, and moreover, that

$$\sum_{l=0}^{k}\alpha_l(1 - \sqrt{\gamma})\|\tilde{p}^\pi - p_l\|_\pi$$

converges.

We now induct on communicating classes, based on the maximal length of path in the directed acyclic graph from the communicating class to a leaf. For any such communicating class $K$, by Perron-Frobenius theory there exists a probability distribution $\kappa$ supported on $K$, with the property that

$$\kappa P^\pi = \lambda\kappa + (1-\lambda)\kappa',$$

with $\lambda \in [0,1)$, and $\kappa'$ a probability distribution supported solely on communicating classes $K'$ for which, by inductive hypothesis, we have $p_k(x) \to \tilde{p}^\pi(x)$ for all $x \in K'$. Now, following exactly the steps of the proof of Proposition 5.3, we obtain

$$\|T^\pi p - T^\pi q\|_\kappa^2 \leq \lambda\gamma\|p-q\|_\kappa^2 + (1-\lambda)\|p-q\|_{\kappa'}^2,$$

where semi-norm $\|\cdot\|_\kappa$ is defined by $\|p\|_\kappa = \sum_{x\in\mathcal{X}} \kappa(x)\|p(x)\|$, and similarly for $\|\cdot\|_{\kappa'}$.

Using the inequality above with $p = \tilde{p}^\pi$ and $q = p_k$, and from the expansion

$$\|\tilde{p}^\pi - T^\pi p_k\|_\kappa^2 = \|\tilde{p}^\pi - p_k\|_\kappa^2 + \|p_k - T^\pi p_k\|_\kappa^2 + 2\langle \tilde{p}^\pi - p_k, \mathbf{C}^\top\mathbf{C}(p^{\phi_k} - T^\pi p_k)\rangle_\kappa,$$

where $\langle u,v\rangle_\kappa = \sum_{x\in\mathcal{X}} \kappa(x)\langle u(x),v(x)\rangle$ $(u,v\in\mathbb{R}^{\mathcal{X}\times m})$ denotes the $\kappa$-weighted inner-product, we obtain

$$\|\tilde{p}^\pi - p_k\|_\kappa^2 + \|p_k - T^\pi p_k\|_\kappa^2 + 2\langle \tilde{p}^\pi - p_k, \mathbf{C}^\top\mathbf{C}(p^{\phi_k} - T^\pi p_k)\rangle_\kappa \leq \lambda\gamma\|\tilde{p}^\pi - p_k\|_\kappa^2 + (1-\lambda)\|\tilde{p}^\pi - p_k\|_{\kappa'}^2,$$

which, with some rearranging, yields

$$\langle \tilde{p}^\pi - p_k, \mathbf{C}^\top\mathbf{C}(p_k - T^\pi p_k)\rangle_\kappa \leq \frac{1}{2}\left((\lambda\gamma - 1)\|\tilde{p}^\pi - p_k\|_\kappa^2 + (1-\lambda)\|\tilde{p}^\pi - p_k\|_{\kappa'}^2 - \|p_k - T^\pi p_k\|_\kappa^2\right).$$

Now, defining

$$L^\kappa(\phi) = \sum_{x\in\mathcal{X}} \kappa(x)\mathrm{KL}(\tilde{p}^\pi(x)\,||\,p^\phi(x)),$$

and writing $L_k^\kappa = L^\kappa(\phi_k)$, we can compute that

$$\begin{aligned}
\mathbb{E}_k[L_{k+1}^\kappa] &\leq L_k^\kappa + \alpha_k\langle\nabla L^\kappa(\phi_k), \mathbf{C}^\top\mathbf{C}(T^\pi p_k - p_k)\rangle + \alpha^2 B \\
&= L_k^\kappa + \alpha_k\langle\tilde{p}^\pi - p_k, \mathbf{C}^\top\mathbf{C}(p_k - T^\pi p_k)\rangle_\kappa + \alpha^2 B \\
&\leq L_k^\kappa + \alpha^2 B + \alpha_k\frac{\lambda\gamma - 1}{2}\|\tilde{p}^\pi - p_k\|_\kappa^2 + \alpha_k\frac{1-\lambda}{2}\|\tilde{p}^\pi - p_k\|_{\kappa'}^2.
\end{aligned}$$

By the induction hypothesis, $\sum_{l=0}^k \alpha_l\|\tilde{p}^\pi - p_l\|_{\kappa'}^2$ converges, and so we may apply the supermartingale convergence theorem as above to conclude that $\sum_{l=0}^k \alpha_l\|\tilde{p}^\pi - p_l\|_\kappa^2$ converges, and that $\|\tilde{p}^\pi - p_k\|_\kappa \to 0$, completing the inductive step and the proof. □

# D. Proofs of results from Section 6

## D.1. Proof of Proposition 6.1

Prior to the proof of Proposition 6.1, we use the following supplementary results.

**Proposition D.1.** *(Meyn, 2022, Theorem 8.1, Proposition 8.10) Let $f : \mathbb{R}^d \to \mathbb{R}^d$, and for an initialization $\theta_0 \in \mathbb{R}^d$, consider the sequence of iterates $(\theta_n)_{n\geq0}$ defined by the iterative rule*

$$\theta_{n+1} = \theta_n + \alpha_{n+1}(f(\theta_n) + \varepsilon_n),$$

*with $(\varepsilon_n)_{n\geq0}$ a mean-zero martingale noise sequence, such that the distribution of $\varepsilon$ depends only on $\theta_n$, and $\alpha_n = \alpha/n^\beta$. If (i) $\partial_t\vartheta_t = f(\vartheta_t)$ has a globally asymptotically stable equilibrium $\theta^*$, (ii) $\theta_n \to \theta^*$ almost surely, (iii) $f$ is Lipschitz continuous, (iv) $\nabla f(\theta^*)$ is Hurwitz, then the scaled error $\alpha_n^{-1/2}(\theta_n - \theta^*)$ converges in distribution to $\mathcal{N}(0,\Sigma)$, where $\Sigma$ is a positive-definite solution of the Lyapunov equation*

$$A\Sigma + \Sigma A^\top = -\Sigma_\Delta,$$

*where $\Sigma_\Delta$ is the covariance of the noise $\varepsilon_n$ at $\theta^*$ and $A = \nabla f(\theta^*)$.*

**Proposition D.2.** *Let $g : \mathbb{R}^d \to \mathbb{R}^k$ be a mapping such that the Jacobian $J_g$ is continuous in a neighbourhood of $\mu \in \mathbb{R}^d$. If $X_n$ is a sequence of $d$-dimensional random vectors such that for some $\beta \in [0, 1]$, $n^\beta(X_n - \mu) \xrightarrow{d} \mathcal{N}(0, \Sigma)$, then*

$$n^\beta(g(X_n) - g(\mu)) \xrightarrow{d} \mathcal{N}(0, J_g \Sigma J_g^\top).$$

*Proof.* This proof closely that of Theorem 7 of Ferguson (1996), but adapted to the scaling $n^\beta$ instead of $\sqrt{n}$. We note that we have $X_n \xrightarrow{d} \mu$. Then if there exists $\delta > 0$ such that $g$ is continuous on the set $\{x \in \mathbb{R}^d : \|x - \mu\|_2 < \delta\}$, for $x$ such that $\|x - \mu\|_2 < \delta$ we have

$$g(x) = g(\mu) + \int_0^1 J_g(\mu + v(x - \mu)) \, dv \, (x - \mu).$$

So for $n$ such that $\|X_n - \mu\| < \delta$,

$$n^\beta(g(X_n) - g(\mu)) = n^\beta \int_0^1 J_g(\mu + v(X_n - \mu)) \, dv \, (x - \mu).$$

Since $X_n \xrightarrow{d} \mu$, $\mathbb{P}(\|X_n - \mu\| < \delta) \to 1$ and $\int_0^1 J_g(\mu + v(X_n - \mu)) \, dv \to J_g(\mu)$, so $n^\beta(g(X_n) - g(\mu)) \xrightarrow{d} J_g(\mu)Z$, where $Z \sim \mathcal{N}(0, \Sigma)$. From this we note that $\mathrm{Dist}(J_g(\mu)Z) = \mathcal{N}(0, J_g \Sigma J_g^\top)$, and we are complete. $\square$

**Lemma D.3.** *Suppose the iterates $(\phi_k^{\mathrm{CMC}})_{k \geq 0}$ and $(\phi_k^{\mathrm{PCMC}})_{k \geq 0}$ were produced using step size $\alpha_k = \alpha_0 k^{-\beta}$ for $\beta \in (1/2, 1)$ and $\tilde{p}_i^\pi(x) > 0$ for each $x, i \in \mathcal{X} \times [m]$. Then we have that there exists unique fixed points $\phi_\star^{\mathrm{CMC}}$ and $\phi_\star^{\mathrm{PCMC}}$ such that have*

$$k^\beta(\phi_k^{\mathrm{CMC}}(x) - \phi_\star^{\mathrm{CMC}}(x)) \xrightarrow{d} \mathcal{N}(0, \Sigma_{\mathrm{CMC}})$$

*and*

$$k^\beta(\phi_k^{\mathrm{PCMC}}(x) - \phi_\star^{\mathrm{PCMC}}(x)) \xrightarrow{d} \mathcal{N}(0, \Sigma_{\mathrm{PCMC}}),$$

*where $\Sigma_{\mathrm{CMC}} = \frac{1}{2}(I - \frac{1}{m}\mathbf{1}\mathbf{1}^\top)$ and $\Sigma_{\mathrm{PCMC}} = \frac{1}{2}(C^\top C - \mathbf{1}\mathbf{1}^\top)$.*

*Proof.* We begin by deriving the fixed points $\phi_\star^{\mathrm{CMC}}$ and $\phi_\star^{\mathrm{PCMC}}$. To begin, we note that from the assumption on full support of $\tilde{p}^\pi$ at each state, the function $\tilde{\theta}(x) = \log(\tilde{p}^\pi(x))$ satisfies $\mathrm{softmax}(\tilde{\theta}(x)) = \tilde{p}^\pi(x)$ for each $x$. Furthermore, $\mathrm{softmax}(\tilde{\theta}(x) + c\mathbf{1}) = \tilde{p}^\pi(x)$ for any $c$, and this construction contains all logits satisfying this condition. We write $\Phi = \{\theta : \mathcal{X} \to \mathbb{R} : \exists c \in \mathbb{R}. \, \theta(x) = \tilde{\theta}(x) + c\mathbf{1}\}$ for this set of optimal logits, and note that each $\phi \in \Phi$ is a fixed point for both CMC and PCMC. We now show that CMC and PCMC always converge to a unique point in this set, regardless of the given sequence of updates.

The CMC update at each step and state is orthogonal to $\mathbf{1}$, as it consists of a difference of probability vectors. This gives us that beginning from $\phi_0^{\mathrm{CMC}}$, for any $k \geq 0$ we have $\mathbf{1}^\top(\phi_0^{\mathrm{CMC}}(x) - \phi_k^{\mathrm{CMC}}(x)) = 0$, for any $x \in \mathcal{X}$. From this we can conclude that $\phi_\star^{\mathrm{CMC}}$ is the unique element of $\Phi$ such that $\mathbf{1}^\top(\phi_\star^{\mathrm{CMC}}(x) - \phi_0^{\mathrm{CMC}}(x)) = 0$ for all $x \in \mathcal{X}$, and we remark that this is entirely dictated by $\phi_0^{\mathrm{CMC}}$. The PCMC updates also maintain an invariant in their updates: in particular the updates do not change the $m$th element of the logit (this can be seen as $(C^\top C e)_m = 0$ for any $e \in \mathbf{1}^\perp$). Thus $\phi_\star^{\mathrm{PCMC}}$ is the unique element of $\Phi$ such that $(\phi_\star^{\mathrm{CMC}}(x))_m = \phi_0^{\mathrm{CMC}}(x)_m$ for all $x \in \mathcal{X}$.

We may now apply Proposition D.1, since CMC converges as stochastic gradient descent on a convex objective with bounded noise, and the convergence of PCMC is guaranteed as a special case of the analysis carried out for PKL-CTD (with discount 0) in Theorem 5.5. The existence of a Lyapunov function for both systems guarantees the global asymptotic stability required as well as the Hurwitz property for the Jacobian at the fixed point. Using Proposition D.1, it now remains to find solutions to the Lyapunov equation of each method. Before applying the lemma, we pause to ensure that all of its conditions are met. The previous arguments gave the unique equilibrium point for both CMC and PCMC, and the remaining assumptions are met if the methods converge. We note that CMC converges as it corresponds to stochastic gradient descent on a convex loss function, and we also have that PCMC converges as it corresponds to PCTD with a discount factor of 0, which we have convergence for (Theorem 5.5).

Now applying Proposition D.1 to CMC, this corresponds to solving

$$J(\tilde{p}^\pi(x))\Sigma + \Sigma J(\tilde{p}^\pi(x)) = J(\tilde{p}^\pi(x))$$

at each state $x \in \mathcal{X}$. As $J(\tilde{p}^\pi(x))$ is not full rank (the vector $\mathbf{1}$ is an eigenvector with eigenvalue 0), there is a 1-dimensional space of solutions. However we also know that the CMC update is orthogonal to $\mathbf{1}$, so the asymptotic variance is the unique solution $\Sigma$ of the Lyapunov equation subject to $\mathbf{1}^\top \Sigma \mathbf{1} = 0$. We now show that this matrix $\Sigma$ is given by $\frac{1}{2}\left(I - \frac{1}{m}\mathbf{1}\mathbf{1}^\top\right)$. To see this we can note that $\mathrm{Col}(\mathbf{1}\mathbf{1}^\top) = \mathrm{span}(\mathbf{1}) = \mathrm{Ker}(J(\tilde{p}^\pi(x)))$, so that $J(\tilde{p}^\pi(x))\Sigma = \frac{1}{2}J(\tilde{p}^\pi(x))$. We also note that as $J(\tilde{p}^\pi(x))$ and $\mathbf{1}\mathbf{1}^\top$ are both symmetric we have that $\mathbf{1}^\top J(\tilde{p}^\pi(x)) = 0$, and hence $\Sigma J(\tilde{p}^\pi(x)) = \frac{1}{2}J(\tilde{p}^\pi(x))$. Combining these we have that

$$J(\tilde{p}^\pi(x))\Sigma + \Sigma J(\tilde{p}^\pi(x)) = J(\tilde{p}^\pi(x)),$$

as desired.

We next turn to PCMC. This now corresponds to solving

$$C^\top C J(\tilde{p}^\pi(x))\Sigma + \Sigma J(\tilde{p}^\pi(x))C^\top C = C^\top C J(\tilde{p}^\pi(x))C^\top C.$$

As in the previous case, this equation admits a 1-dimensional subspace of solutions, however from the assumption above we know that we must have 0 covariance in the bottom right element of $\Sigma$, as there is no change in this logit. We can then find that $\frac{1}{2}(C^\top C - \mathbf{1}\mathbf{1}^\top)$ satisfies this condition, and we can verify that this satisfies the Lyapunov equation. $\qquad\square$

**Lemma D.4.** *Under the same assumptions as Lemma D.3, we have*

$$k^\beta(p^{\theta_k}(x) - \hat{p}^\pi(x)) \xrightarrow{d} \mathcal{N}\left(0, \frac{1}{2}J(\tilde{p}^\pi)^2\right),$$

*and*

$$k^\beta(p^{\vartheta_k}(x) - \hat{p}^\pi(x)) \xrightarrow{d} \mathcal{N}\left(0, \frac{1}{2}J(\tilde{p}^\pi)C^\top C J(\tilde{p}^\pi)\right),$$

*where we use the shorthand $J(p) = \mathrm{diag}(p) - pp^\top$.*

*Proof.* This follows from the convergence result of Lemma D.3 and Proposition D.2. In particular we are applying the function $\phi(x) \mapsto p^\phi(x)$, which has Jacobian $J(p^\phi(x))$. We can also note that $\mathbf{1}$ is in the kernel of $J(p^\phi(x))$, so that

$$J(p^\phi(x))(I - \frac{1}{m}\mathbf{1}\mathbf{1}^\top)J(p^\phi(x))^\top = J(p^\phi(x))^2$$

and

$$J(p^\phi(x))(C^\top C - \mathbf{1}\mathbf{1}^\top)J(p^\phi(x))^\top = J(p^\phi(x))C^\top C J(p^\phi(x)).$$

With this, we are complete. $\qquad\square$

**Proposition 6.1.** *Suppose the iterates $(V_k)_{k\geq 0}$, $(\phi_k^{\mathrm{CMC}})_{k\geq 0}$, and $(\phi_k^{\mathrm{PCMC}})_{k\geq 0}$ were produced using step size $\alpha_k = \alpha_0 k^{-\beta}$ for $\beta \in (1/2, 1)$. Further suppose that $\breve{p}_i^\pi(x) > 0$ for each $x, i \in \mathcal{X} \times [m]$. Then we have that*

$$k^{\beta/2}(V_k(x) - V^\pi(x)) \xrightarrow{d} \mathcal{N}(0, \sigma^2(x))$$
$$k^{\beta/2}(V_{\phi_k}^{\mathrm{CMC}}(x) - V^\pi(x)) \xrightarrow{d} \mathcal{N}\left(0, \frac{1}{2}z^\top J(\breve{p}^\pi(x))^2 z\right)$$
$$k^{\beta/2}(V_{\phi_k}^{\mathrm{PCMC}}(x) - V^\pi(x)) \xrightarrow{d} \mathcal{N}\left(0, \frac{1}{2}u(x)^\top u(x)\right),$$

*where $J(p) = \mathrm{diag}(p) - pp^\top$, $u(x) = CJ(\breve{p}^\pi(x))z$ and $\sigma^2$ is the variance of $\eta^\pi(x)$.*

*Proof.* This follows from the results of Lemma D.4 and Proposition D.2, as we are passing the converging probabilities of Lemma D.4 through the map $p \mapsto p^\top z$, and the effect on the limiting distribution from this transformation is given by Proposition D.2. $\qquad\square$

## D.2. Proof of Proposition 6.2

**Proposition 6.2.** *Suppose $p = (p_i)_{i=1}^m$ is a collection of probabilities associated to the locations $z = (z_i)_{i=1}^m$, and let the variance of this categorical distribution be $\sigma^2$. Then there exists $\beta$ depending only on $p$ such that*

$$\beta\sigma^2 < z^\top J(p)^2 z < \sigma^2.$$

*Proof.* Note that $J(p)$ is a positive-semidefinite matrix with all eigenvalues in $[0, 1)$ (we have $x^\top J(p)x \geq 0$ for any $x$ as it is the variance of a random variable distributed as $\sum_i p_i \delta_{x_i}$, and its maximal eigenvalue must be strictly less than 1 as its trace is $\sum_i p_i(1 - p_i) \leq \frac{m-1}{m}$). Writing $0 = \lambda_1 \leq \cdots \leq \lambda_m < 1$ for the eigenvalues of $J(p)$, we note that $\lambda_1^2, \ldots, \lambda_m^2$ are the eigenvalues of $J(p)^2$, and $J(p), J(p)^2$ share the same respective eigenvectors as $J(p)$ is symmetric. Next writing $\tilde{z}_1, \ldots, \tilde{z}_m$ for the coordinates of $z$ in the basis of corresponding eigenvectors of $J(\tilde{p}^\pi)$, we have that

$$z^\top J(p)z = \sum_{i=1}^m \tilde{z}_i^2 \lambda_i,$$

and

$$z^\top J(p)^2 z = \sum_{i=1}^m \tilde{z}_i^2 \lambda_i^2.$$

As each $\lambda_i < 1$, this results in a strictly smaller quantity. We can also note that $z^\top J(p)z \leq \lambda_m \|z\|_2$ and $z^\top J(p)^2 z \geq \lambda_2^2 \|z\|_2$, so that $\frac{z^\top J(p)z}{z^\top J(p)^2 z} \leq \frac{\lambda_m}{\lambda_2^2}$, and we can set $\beta = \frac{\lambda_2^2}{\lambda_m}$ to satisfy the statement in the proposition. $\square$

## D.3. Proof of Proposition 6.3

**Proposition 6.3.** *Suppose $\nu$ is a probability measure whose support is contained in $[a, b]$, and $\mathrm{Var}_{Z \sim \nu}(Z) = \sigma^2$. Let $a = z_1, \ldots, z_m = b$ be $m$ equally spaced points, Then we have*

$$\mathrm{Var}_{Z \sim \Pi_C \nu}(Z) = \sigma^2 + E(\nu) \geq \sigma^2,$$

*where $0 \leq E(\nu) \leq \frac{(b-a)^2}{4(m-1)^2}$ is a quantity capturing the amount of projection in the map $\nu \mapsto \Pi_C \nu$.*

*Proof.* Let $Z \sim \nu$, then by the assumption on the support of $\nu$ we have that $z_1 \leq Z \leq z_m$ almost surely. Our goal is to introduce a jointly distributed random variable $E$ distributed as the difference between $Z$ and the Cramér projection of $Z$. We define $l(Z) = \max\{z_i : z_i \leq Z\}$, and $u(Z) = \min\{z_i : z_i \geq Z\}$. If $l(Z) = u(Z)$ then $Z$ is already on a support point of $\Pi_C \nu$, and in this case we set $E$ to be 0 with probability 1. In the case that $l(Z) < u(Z)$, we set $E$ to take value $(Z - l(Z))$ with probability $(u(Z) - Z)/(u(Z) - l(Z))$, and $(u(Z) - Z)$ with probability $(Z - l(Z))/(u(Z) - l(Z))$. Then we have that $Z + E$ takes value $l(Z)$ with probability $(u(Z) - Z)/(u(Z) - l(Z))$ and $u(Z)$ with probability $(Z - l(Z))/(u(Z) - l(Z))$, that is, $Z + E$ is distributed according to $\Pi_C \nu$.

We can note that $\mathbb{E}[E \mid Z] = 0$, which gives us that $E$ and $Z$ are uncorrelated. We can also consider $\mathrm{Var}(E \mid Z)$, which therefore reduces to computing $\mathbb{E}[E^2 \mid Z]$. When $Z$ lies on the grid, this quantity is trivially 0, and otherwise we have

$$\begin{aligned}
\mathbb{E}\left[E^2 \mid Z\right] &= (Z - l(Z))^2 \cdot \frac{(u(Z) - Z)}{(u(Z) - l(Z))} + (u(Z) - Z)^2 \cdot \frac{(Z - l(Z))}{(u(Z) - l(Z))} \\
&= \frac{(Z - l(Z))(u(Z) - Z)\Big((u(Z) - Z) + (Z - l(Z))\Big)}{(u(Z) - l(Z))} \\
&= (Z - l(Z))(u(Z) - Z).
\end{aligned}$$

We next set $U = Z + E$, and write out

$$
\begin{aligned}
\operatorname{Var}(U) &\overset{(a)}{=} \operatorname{Var}(Z) + \operatorname{Var}(E) \\
&= \operatorname{Var}(Z) + \mathbb{E}\left[\operatorname{Var}(E \mid Z)\right] + \operatorname{Var}\left(\mathbb{E}\left[E \mid Z\right]\right) \\
&= \operatorname{Var}(Z) + \mathbb{E}\left[\operatorname{Var}(E \mid Z)\right] \\
&= \operatorname{Var}(Z) + \sum_{i=1}^{m-1} \int_{z_i}^{z_{i+1}} \operatorname{Var}(E \mid Z = z)\, \mathbb{P}(dz) \\
&= \operatorname{Var}(Z) + \sum_{i=1}^{m-1} \int_{z_i}^{z_{i+1}} (z - l(z))(u(z) - z)\, \mathbb{P}(dz),
\end{aligned}
$$

where (a) follows from the deduction above that $Z$ and $E$ are uncorrelated. This quantity is minimized when $\nu$ is supported on $\{z_1, \ldots, z_m\}$, as the additional variance term is 0. The quantity $(z-l(z))(u(z)-z)$ is maximized when $z = (u(z)+l(z))/2$, meaning $z$ is concentrated on the midpoint of $z_i$ and $z_{i+1}$, that is, the case when $\nu = \sum_{i=1}^{m-1} p_i \delta_{(z_i+z_{i+1})/2}$, and the amount of projection is maximal. In this case, the additional variance incurred is given by

$$
\begin{aligned}
\sum_{i=1}^{m-1} p_i \frac{(u(z)-l(z))^2}{4} &= \sum_{i=1}^{m-1} p_i \frac{(b-a)^2}{4(m-1)^2} \\
&= \frac{(b-a)^2}{4(m-1)^2}.
\end{aligned}
$$

$\square$

## D.4. Proof of Proposition 6.5

**Proposition 6.5.** *If the step sizes $(\alpha_k)_{k \geq 0}$ are given by $\alpha_k = \alpha_0 k^{-\beta}$, we have that*

$$
k^{\beta/2}(V_{\phi_k}^{\mathrm{PCTD}}(x) - V^\pi(x)) \overset{d}{\to} \mathcal{N}\left(0, b_x^\top\, \Sigma_{\mathrm{PCTD}}(x)\, b_x\right),
$$

*where $b_x = J(\tilde{p}^\pi(x))z$ and $\Sigma_{\mathrm{PCTD}}(x)$ is the unique solution $\Sigma$ of the Lyapunov equation*

$$
\mathbf{A}(T^\pi - I)J(p)\Sigma + \Sigma J(p)((T^\pi)^\top - I)\mathbf{A} + \mathbf{A}J(p)\mathbf{A} = 0
$$

*subject to $\Sigma_{mm} = 0$, where we write $p = \tilde{p}^\pi(x)$, $\mathbf{A} = \mathbf{C}^\top \mathbf{C}$, and $T^\pi$ as introduced in Proposition 4.1.*

*Proof.* We may apply Proposition D.1. Our earlier Theorem 5.5 guarantees the conditions of the result hold, thanks to the establishing of a Lyapunov function; the existence of the Lyapunov function establishes global asymptotic stability of the equilibrium, and the expected update function is clearly Lipschitz (since as noted earlier, the softmax function is Lipschitz), and the Lyapunov function also guarantees that the the Jacobian at the equilibrium point is Hurwitz. We thus obtain the stated Lyapunov equation for the term $\Sigma_{\mathrm{PCTD}}$. This equation will have a 1-dimensional subspace of solutions due to the rank-deficiency of $J(p)$, however the PKL-CTD updates do not modify the final logit, so we must have the bottom right entry of the covariance matrix be 0. This constraint uniquely identifies the matrix in the subspace of solutions as the subspace of solutions is along the span of $\mathbf{1}\mathbf{1}^\top$. This then combined with Proposition D.2 to reflect the effect of the map $\phi \mapsto z^\top p^\phi$ completes the statement. $\square$

# E. Related work

Categorical approaches to tasks that are traditionally modeled as regression have been studied in a variety of settings more generally. Stewart et al. (2023) study the effects on neural network representations, and Lyle et al. (2024) study the effects on network plasticity.

Beyond categorical distributional reinforcement learning (Bellemare et al., 2017), a variety of categorical approaches to reinforcement learning have been considered in the literature, including two-hot regression targets, and various versions

of one-step distributional RL (Achab, 2020; Schrittwieser et al., 2020; Hoffman et al., 2020; Achab et al., 2023; Hafner et al., 2023). In particular, Farebrother et al. (2024) study the empirical benefits of categorical algorithms for reinforcement learning, finding a combination of one-step categorical distributional reinforcement learning (Achab, 2020; Achab et al., 2023) and implicitly injected Gaussian noise to smooth the resulting loss (Imani & White, 2018; Imani et al., 2024) particularly effective, producing a novel reinforcement learning algorithm that was demonstrated to be performant in a wide variety of applications.

The Cramér-variant of CTD was first analysed by Rowland et al. (2018), who proved its convergence to the fixed point of the categorical dynamic programming algorithm. Boeck & Heitzinger (2022) then developed a variant of this algorithm that incorporates ideas from speedy Q-learning (Azar et al., 2011); both Boeck & Heitzinger (2022) and Peng et al. (2024) obtain finite-time sample complexity bounds on Cramér-based CTD algorithms. An analysis of a preconditioned variant of Cramér-CTD was performed by Peng et al. (2025).

Rowland et al. (2023) study the efficacy of quantile temporal-difference learning (Dabney et al., 2018; Rowland et al., 2024), a complementary approach to distributional reinforcement learning, for value-function estimation. Several recent works also analyze the complementary class of likelihood-maximization-based methods for distributional reinforcement learning, both from the perspective of regret minimization (Wang et al., 2023; 2024b; Ayoub et al., 2024) and sample-efficient policy optimization and evaluation (Ayoub et al., 2024; Wu et al., 2023; Wang et al., 2024a).

## F. Additional experimental results

We perform ablations over potential causes for the occasional underperformance of TD: synchronous vs asynchronous updates, boundedness of the value estimates, and boundedness of the regression targets. The experimental suite in Section 7 uses asynchronous updates for TD, KL-CTD, and PKL-CTD, which may cause an increased difference in the number of updates across states. We present the experimental suite using synchronous updates in Figure 10. The second potential cause is boundedness of the value estimate: due to the nature of the categorical parametrization, the categorical value estimate at each state is bounded in $[z_1, z_m]$. To see if this has a material effect on the relative performance of the algorithms, we consider a variant of TD learning where the value estimate at each state is clipped to $[z_1, z_m]$ after each update. We present the experimental suite under these updates in Figure 11. The third potential cause is that the categorical target distributions are always bounded in $[z_1, z_m]$, which may be contributing to a regularization effect, especially with heavy-tailed rewards. To test this we consider a variant of TD learning where the TD target is clipped to $[z_1, z_m]$ before being used in the update step, and present the experimental suite using these updates in Figure 12.

Across all of the above considered ablations, the relative performance of TD, KL-CTD, and PKL-CTD are fairly robust, suggesting that in the settings where KL-CTD or PKL-CTD outperform TD, this is more likely to be a result of the underlying algorithmic differences, rather than an effect of the relative number of updates per state or clipping effects of the categorical parametrization.

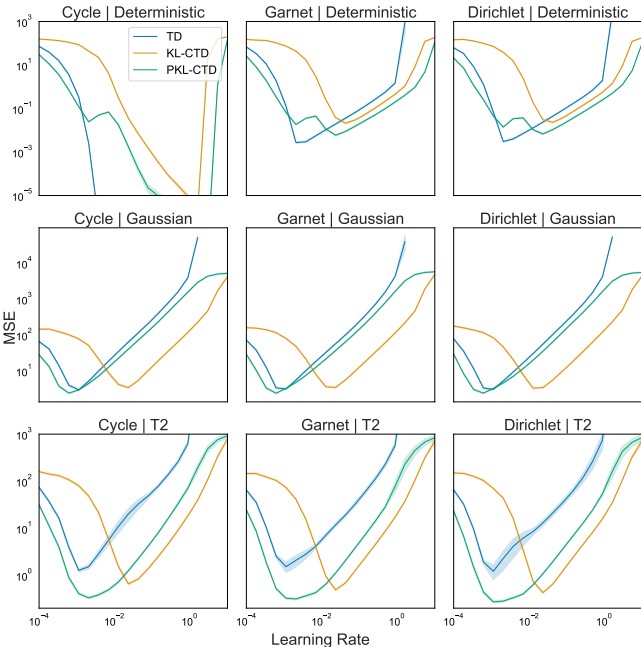

*Figure 10.* Experimental suite where each algorithm is run synchronously.
.

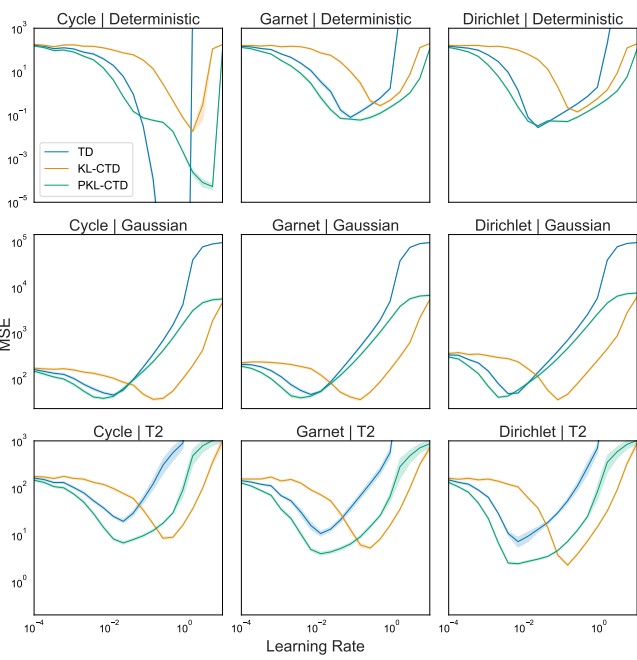

*Figure 11.* Experimental suite where TD estimates are clipped to $[z_1, z_m]$ after each update.

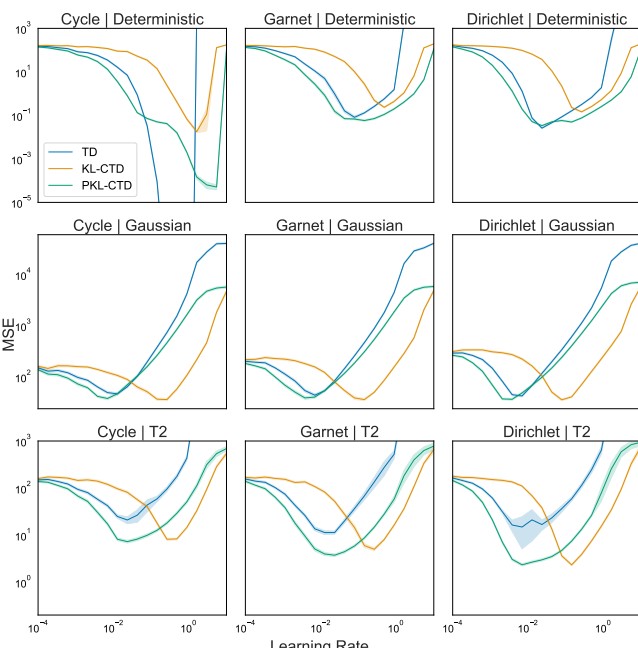

*Figure 12.* Experimental suite where TD targets are clipped to $[z_1, z_m]$ prior to each update.

# G. Experimental details

In this section, we collect full details for the experiments presented in the main paper. For all generated figures, the error bars represent $\pm 2$ times the sample standard error of the mean.

## G.1. Details for Figure 2

The MDP considered is a two-state MDP, with a single action $a$ and discount factor 0. Writing the states as $x, y$, we have $\mathbb{P}(x \mid x, a) = 1$ and $\mathbb{P}(y \mid y, a) = 1$, that is each state transitions to itself. The reward distributions are given as $\mathcal{R}(x, a) = \delta_0$, $\mathcal{R}(y, a) = \frac{1}{2}(\delta_{-10} + \delta_{10})$. Cramér-CTD and KL-CTD are both run using 40 atoms uniformly spaced on $[-30, 30]$, a learning rate of $4 \cdot 10^{-3}$ was used for TD and Cramér-CTD, and a learning rate of $1 \cdot 10^{-1}$ was used for KL-CTD.

## G.2. Details for Figure 3

We use a two-state MDP with a single action, deterministic rewards, and a discount factor of 0.7. The transition matrix is sampled from a $\text{Dirichlet}(1, \ldots, 1)$ distribution, and the reward means are sampled independently from $\mathcal{N}(0, 1)$. KL-CTD was run for 5,000 iterations with 50 atoms uniformly spaced on $[-10, 10]$ and a learning rate of 0.5.

## G.3. Details for Figure 4

We use a single-state MDP with a single action and discount factor 0. The return from this state is sampled from $U([-10, 10])$. We use CTD with 10 atoms whose locations are uniformly spaced across $[-10, 10]$ with a step size sequence given by $\alpha_k = k^{-0.55}$.

## G.4. Details for Figure 5

We consider an MDP with 10 states, transitions sampled from a $\text{Dirichlet}(1, \ldots, 1)$ distribution, deterministic rewards whose means are sampled from $U([0, 1])$, and discount factor 0.5. We set the number of atoms to be 20, and uniformly space these atoms across $[0, z_m]$. The final atom location $z_m$ itself is swept over 100 uniformly spaced points in $[0.1, 10]$.

## G.5. Details for Figure 6

We consider an MDP with 20 states, transitions sampled from a $\text{Dirichlet}(1, \ldots, 1)$ distribution, deterministic rewards whose means are sampled from a standard normal $\mathcal{N}(0, 1)$ distribution, and discount factor 0.9. We sweep over the number of atoms, $m$, in the set $[4, 10, 16, 50, 100, 200]$. For each choice of $m$, we set the locations to be uniformly spaced on $[-10, 10]$. We swept over learning rates uniformly-spaced in log-space over the range $[1 \cdot 10^{-4}, 10]$. For the scaled learning rates, we scale each learning rate by $\sqrt{m}$, so that a learning rate of $\alpha$ corresponds to an *effective* learning rate of $\alpha\sqrt{m}$.

## G.6. Details for Figure 7

All MDPs considered have 20 states and a discount factor of 0.7. All reward functions have their means sampled from a standard normal $\mathcal{N}(0, 1)$ distribution. All methods are initialized so that their initial value estimates are 0, and CTD and PCTD use 50 bins uniformly spaced on $[-100, 100]$. We sweep over learning rates uniformly-spaced in log-space over the range $[10^{-4}, 10]$. We now detail each type of transition and reward structure.

**Transition dynamics**

- Cycle domain: We use a deterministic cycle transition structure over the states.
- Garnet domain: We sample a sparse Garnet MDP transition structure (Archibald et al., 1995).
- Dirichlet domain: Each row of the transition matrix is sampled i.i.d. from a $\text{Dirichlet}(1, \ldots, 1)$ distribution.

**Reward distributions**

- Deterministic distributions: The distributions are Dirac distributions centred at the means.
- Gaussian distributions: The reward distributions are Gaussian with the given means with variance 1.
- $t_2$ distributions: The reward distributions are shifted $t_2$ distributions to maintain the specified reward means.

