# OpenReview forum: "Categorical Distributional Reinforcement Learning with Kullback-Leibler Divergence: Convergence and Asymptotics"
_ICML.cc/2025/Conference — ICML 2025 poster_

### Official Review · Reviewer_tUof · 2025-02-20

**Overall Recommendation:** 3

**Summary:**

This paper analyzed categorical TD learning with KL loss in the tabular setting. They also proposed a preconditioned version of the algorithm and provided an asymptotic normal analysis. They also conducted experiments to verify the theoretical results.

**Claims And Evidence:**

Yes

**Essential References Not Discussed:**

No.

**Experimental Designs Or Analyses:**

Yes

**Methods And Evaluation Criteria:**

Yes

**Other Comments Or Suggestions:**

Some notations are used without being defined., such as \|... \|_{\pi}, d^\pi.
There is typo in the equation in Line 225-228.

**Other Strengths And Weaknesses:**

Strength:
1. The motivation is good. On the one hand, the KL loss (instead of the Cramer loss) is adopted by many practical algorithms, and studying the performance under this loss function helps in understanding real-world algorithms. On the other hand, it also inspires the development of distributional reinforcement learning algorithms that utilize other probabilistic metrics as loss functions.
2. It is a good idea to use preconditioning to improve KL-CTD and guarantee convergence.
3. For readers, it is quite clear to use the asymptotic variance to compare the performance of algorithms in the simple Monte-Carlo and tabular settings.

Weakness:
1. Although the motivation is quite good, and there are also experimental and some theoretical evidences to illustrate the advantages of using the KL loss compared with the Cramer loss, considering that this paper focuses on the simplified tabular setting, I think it would be better to have theoretical evidence demonstrating that, in terms of learning distributions (not just the means, i.e. value functions in Section 6), the KL loss has advantages over the Cramer loss. For example, you can provide the asymptotic normality of the vector p_k^{Cramer-CMC} and p_k^{KL-CMC}
2. The paper mentioned that the KL loss is used in many practical algorithms. If there could be experimental comparisons between the KL loss and the Cramer loss (for example, see linear function approximation algorithm in [1]) under the function approximation setting, I believe the paper would be more complete.
[1] Lyle, C., Bellemare, M. G., and Castro, P. S. A comparative analysis of expected and distributional reinforcement
learning. In Proceedings of the AAAI Conference on Artificial Intelligence, 2019.

**Questions For Authors:**

No

**Relation To Broader Scientific Literature:**

This paper analyzes categorical TD learning with KL loss, which are used in many deep distributional reinforcement learning algorithms.

**Theoretical Claims:**

Yes

---

> ### Author Rebuttal · Authors · 2025-04-01
>
> We thank the reviewer for taking the time and effort in reviewing our paper, and we are pleased that they found our motivation compelling, appreciated our use of preconditioning to obtain a convergence guarantee, and found our use of asymptotic variance for the analysis to be a clear choice.
>
>
> **Comparison between Cramér and KL losses for learning distributions**
>
> We thank the reviewer for this suggestion, as we agree that an asymptotic comparison of the probability vectors learnt by the Cramér and KL losses is a valuable comparison to make. We report this result in a new proposition, which we state below.
> Let us define the iterates $p_k^{\text{Cram\’er-CMC}}$ as follows: $p_{k+1}^{\text{Cram\’er-CMC}} = p_k^{\text{Cram\’er-CMC}} + \alpha_k (h(G) - p_k^{\text{Cram\’er-CMC}})$. Then under the assumptions of Lemma D.3., we have that
>
> $$k^\beta ( p_{k}^{\text{Cram\’er-CMC}}  - \tilde{p}^\pi)  \overset{d}{\to} \mathcal{N}\left(0, \mathbb{E}_{G}[h(G) h(G)^T] - (\tilde{p}^\pi)(\tilde{p}^\pi)^T\right).$$
>
>
> We also want to emphasize that the goal of our work is not necessarily to argue that the KL version of the algorithm is always superior to the Cramer-based version (and indeed this is not always the case).
> Our main motivation is to study the KL version in its own right, which is the fundamental algorithm underlying the original deep RL implementation of distributional RL (C51; Bellemare et al., 2017).
>
>
>
>
>
> **Experimental comparisons between the KL loss and the Cramer loss under the function approximation setting**
>
> We appreciate that while our focus on the tabular setting potentially seems a limitation due to its simplicity, we want to emphasize that this was a conscious choice in order to develop a core understanding of categorical TD algorithms based on KL gradients, which have not been studied before, without confounding factors that often arise in larger-scale experiments (such as adaptive optimisation, function approximation, target networks, etc.).
>
> We additionally note that the paper suggested by the reviewer performs an empirical comparison of KL-CTD and Cramér-CTD in the function approximation setting: C51 is the deep learning equivalent of KL-CTD, and S51 as used in their paper is the deep learning equivalent of Cramér-CTD.
>
>
> **Other Comments Or Suggestions**
>
> We thank the reviewer for bringing attention to these typos. We will be sure to fix them in the text.
>
>
> **References**
>
> Bellemare, M. G., Dabney, W., and Munos, R. A distributional perspective on reinforcement learning. In International Conference on Machine Learning, 2017.

---

> > ### Comment · Reviewer_tUof · 2025-04-02
> >
> > Thank you for your reply. I think authors has addressed all my questions, and I will keep my positive score.

---

### Official Review · Reviewer_YLKh · 2025-03-08

**Overall Recommendation:** 3

**Summary:**

In this paper, the authors studied the theoretical properties of categorical distributional TD with KL loss. They proposed a preconditioned version of the algorithm called PKL-CTD, proved its asymptotic convergence, and derived the asymptotic distribution of the resulting value estimators. These theoretical results also provide valuable insights for practitioners.

**Claims And Evidence:**

The claims made in the submission are supported by sound theoretical analysis or empirical evidence.

**Essential References Not Discussed:**

No.

**Experimental Designs Or Analyses:**

The experiments are adequate for the purpose of validating the theoretical findings.

**Methods And Evaluation Criteria:**

The proposed methods make sense for the problem at hand.

**Other Comments Or Suggestions:**

I do not see the blue line indicating the MSE of TD learning in Figure 2, I hope the author can fix it in later revisions.

**Other Strengths And Weaknesses:**

Strengths:

* The idea of introducing a preconditioner is concise and intuitive.
* The asymptotic analysis of the value estimator provides a series of valuable insights, for example, the bias-variance tradeoff incurred by the choice of the size of supports.

Weaknesses:

* The asymptotic analysis only focuses on the asymptotic behavior of value estimators in TD and KL-CTD. I think it may also be helpful to compare the asymptotics between Cramer-CTD and KL-CTD.
* As the authors state, "However, most large-scale implementations of categorical temporal-difference learning use a KL loss, rather
than Cramer loss. This is a crucial detail of large-scale implementations, but has not yet been theoretically analyzed." However, it seems that the author only considers the tabular case. I think the authors should at least use some experiments to show that the insights obtained in the tabular case are also valid in the large-scale case with the function approximation techniques applied.

**Questions For Authors:**

Can the authors generalize the asymptotic analysis to estimators of statistical functionals other than the mean? For example, the asymptotic distribution of differentiable statistical functionals can be obtained with a combination of asymptotics of the learned weights and delta method.

**Relation To Broader Scientific Literature:**

This paper presents a thorough discussion of the theoretical properties of categorical distributional TD learning with KL loss. The theoretical findings advance the understanding of the distributional TD learning algorithm and provide valuable insights for practitioners. I think the authors has made a solid contribution to the field of distributional RL.

**Theoretical Claims:**

The theoretical results seem reasonable in this work. I checked most parts of the proof and found them correct.

---

> ### Author Rebuttal · Authors · 2025-04-01
>
> We thank the reviewer for their time and effort in reviewing our paper, and for the helpful feedback and suggestions provided. We are pleased to hear that they found our work to provide valuable insights for practitioners and our idea of preconditioning to be concise and intuitive.
>
> **Asymptotic analysis missing for Cramér-CTD**
>
> We thank the reviewer for pointing out this limitation, and we have expanded the asymptotic results based on their feedback. In particular, we have derived additional results for the asymptotic normality of $p_k^{Cramer-CMC}$, and present this below.
>
> Let us define the iterates $p_k^{\text{Cram\’er-CMC}}$ as follows: $p_{k+1}^{\text{Cram\’er-CMC}} = p_k^{\text{Cram\’er-CMC}} + \alpha_k (h(G) - p_k^{\text{Cram\’er-CMC}})$. Then under the assumptions of Lemma D.3., we have that
>
> $$
> k^\beta ( p_{k}^{\text{Cram\’er-CMC}}  - \tilde{p}^\pi)  \overset{d}{\to} \mathcal{N}\left(0, \mathbb{E}_{G}[h(G) h(G)^T] - (\tilde{p}^\pi)(\tilde{p}^\pi)^T\right).
> $$
>
>
> **Extension of asymptotic analysis to general statistical functionals**
>
> We thank the reviewer for highlighting this as a direction to expand our current results, as we find this to be a nice generalization of our current analysis. We present the result for KL-CMC below, and will include it for TD and Cramér-CMC in the paper.
>
> Suppose $\psi: \mathscr{P}(\mathbb{R}) \to \mathbb{R}^k$ is a statistical functional sketch, and let $J_\psi: \mathbb{R}^m \to \mathbb{R^k}$ be the Jacobian of $\phi \mapsto \psi(p^\phi)$. Then if $J_\psi$ is continuous in a neighbourhood of $\tilde{p}^\pi$, we have
>
> $$
> k^\beta ( \psi(p_{k}^{\text{KL-CMC}})  - \psi(\tilde{p}^\pi))  \overset{d}{\to} \mathcal{N}\left(0, \frac12 J_\psi(\tilde{p}^\pi) \left(\mathrm{diag}(\tilde{p}^\pi) - (\tilde{p}^\pi)(\tilde{p}^\pi)^\top\right) J_\psi(\tilde{p}^\pi)^\top\right).
> $$
>
>
> **Work only considers the tabular case**
>
> We appreciate that while our focus on the tabular setting potentially seems a limitation due to its simplicity, we want to emphasize that this was a conscious choice in order to develop a core understanding of categorical TD algorithms based on KL gradients, which have not been studied before, without confounding factors that often arise in larger-scale experiments (such as adaptive optimisation, function approximation, target networks, etc.).
>
>
> **Missing blue line in Figure 2**
>
> The line for the MSE of TD is not missing, it rather perfectly overlaps with the line of Cramér-CTD, as predicted by the theory of Lyle et al. (2019). This is a key motivation for the work: while the behaviour of Cramer-CTD exactly coincides with classical TD learning, as predicted by Lyle et al. (2019), this shows that there are settings with KL-CTD has distinct, and sometimes superior, performance to these other algorithms, motivating us to understand the theory of this approach better. We will make this more emphasized in the text, and explore other methods of visualization to remove any confusion, such as making the linewidth of the blue curve larger.
>
>
> **References**
>
> Lyle, C., Bellemare, M. G., and Castro, P. S. A comparative analysis of expected and distributional reinforcement learning. In Proceedings of the AAAI Conference on Artificial Intelligence, 2019.

---

> > ### Comment · Reviewer_YLKh · 2025-04-02
> >
> > I want to thank the authors for their responses. I will retain my positive rating of this paper.

---

### Official Review · Reviewer_wwL4 · 2025-03-09

**Overall Recommendation:** 2

**Summary:**

This paper revisited categorical distributional RL and conducted the analysis based on KL divergence instead of the conventional Cramer distance. The authors proposed a variant of the algorithm by preconditioning and showed its convergence. More importantly, the asymptotic normality or asymptotic variance is also provided in estimating the values. Experiments are conducted in some simple environments to demonstrate the theoretical results.

## update after rebuttal

Thank you for the response. This paper has many technical contributions, but I still think it could be largely enhanced. In particular, the authors can consider strengthening the motivation for why we need to fill the gap between Cramer CTD and KL CTD here, beyond only providing some simple motivating examples. The theoretical contributions should be made clearer. Also, I would suggest one or two conclusions be emphasized in the end instead of applying a tree-manner writing style that directly stacks multiple insights. Thus, I keep my rating.

**Claims And Evidence:**

This paper gives me the feeling that many parts are disconnected without presenting one or two clear conclusions, even though each part is involved with rigorous analysis. For example, why do the authors consider the KL variant categorical distributional RL, and what is the gap between it from the original algorithm, such as C51? Why do we then move to a preconditioned variant of an algorithm rather than rethink the properties of real categorial distributional RL algorithms? Furthermore, I understand it takes effort to conduct the convergence and asymptotic analysis in value estimates, but what are the purposes or goals of presenting these results? Do the authors hope to demonstrate an advanced algorithm that can be superior to the original categorical distributional RL or just ``dump’ ’related properties that could be derived? After reading this paper, I am very confused about the main purpose of this paper and what kind of insights or conclusion the readers should gain after the reading.

**Essential References Not Discussed:**

I think the references are relatively self-contained. It is a theory-focused paper that mainly discusses the related work directly linked with the analysis, but it would be better to add more references on the whole distributional RL area.

**Experimental Designs Or Analyses:**

Although this is a theoretical paper, the experiments are still weak from my perspective. For example, Figures 2 and 3 are on a particularly chosen environment, which is less convincing. The results in Figure 4 are appealing, but they are conducted on a very simple one-state environment after I checked the details in the appendix. In addition, in Section 7.2, it seems that the superiority of the proposed algorithms PKL-CTD over KL-CTD and TD highly depends on the step size (learning rate), which is frustrating, especially for practitioners. It is suggested to have experiments on some classical gym control environments, even though it may not be necessary to consider complex environments for a theory-focused paper.

**Methods And Evaluation Criteria:**

What is the motivation to consider categorical distributional RL with KL divergence? Is that because the previous analysis in (Rowland et al. 2018), such as Cramer TD, does not align well with the real practical algorithm? Or is it just because the KL variant algorithm could simplify the theoretical analysis?

The motivation for the study of KL-CTD is not clear in Section 3, given that the environments are particularly chosen. In addition, the original categorical distributional RL may also behave erratically in Figure 3. Why do we not directly study the original algorithm?

I find it difficult to understand the motivation of preconditioning variants. It seems to be mainly from a theoretical perspective.

**Other Comments Or Suggestions:**

What is the gap between KL-based categorical distributional RL and the vanilla algorithm?

**Other Strengths And Weaknesses:**

Please see the comments above.

**Questions For Authors:**

It looks like this is a follow-up paper to Rowland (2018). What is this paper's main novelty or contribution relative to it?

**Relation To Broader Scientific Literature:**

Although I agree that the theoretical analysis looks rigorous for this paper, my concern is that it may not be related to the broader scientific literature. The focus of the paper is a modification of categorical distributional RL, which is one specific distributional RL algorithm. Although this modification with KL divergence simplifies the theoretical analysis, allowing rich convergence and asymptotic normality analysis in value estimates, it may still be specific. I am also looking for some potential theoretical contributions that may be generalized to broader areas. However, I feel that many analyses are based on existing results and are established in a parallel manner. Therefore, I am concerned about whether this paper could be related to broader literature, given that the analytic tools may be a little specific in a variant of categorical distributional RL.

**Theoretical Claims:**

The theoretical claims look fine to me. One question is how to understand the convergence result in this paper based on KL divergence, given that distributional dynamic programming is only non-expansive if we directly employ the KL divergence to measure the distribution distance.

Is $\tilde{p}_i^{\pi}$ above Eq. 8 in $\tilde{\eta}$ the probability after the two-hot transformation?

Does Proposition 5.3 mean that KL-CTD converges to the same fixed point of the original categorical distributional RL algorithm?

It is not clear why $C$ is introduced in Eq.9 and 10. Could any intuitive explanation be provided?

Why consider the asymptotic normality/variance in value estimates in Propositions 6.1 and 6.4? What is the main regularity condition? Is there any potential application of these asymptotic properties?

---

> ### Author Rebuttal · Authors · 2025-04-01
>
> We thank the reviewer for their time and effort they spent in reviewing our paper, and we aim to resolve their questions below.
>
> **Why do the authors consider the KL variant of categorical distributional RL, and what is the gap between it from the original algorithm, such as C51?**
>
> We focus on the KL variant of categorical distributional RL as it is exactly the fundamental algorithm used by C51. The gap between KL-CTD considered here and C51 is due to the use of policy evaluation instead of control, and tabular analysis instead of the deep learning setting.
>
> **The motivation for the study of KL-CTD is not clear in Section 3. Why do we not directly study the original algorithm?**
>
> We motivate KL-CTD in two ways in Section 3:
>
> - Firstly, KL-CTD is the original categorical TD algorithm that C51 makes use of. This in itself is a strong motivation for its study.
> - Our examples in Figure 2 and 3 serve as further motivation, showing that even in tabular settings, without neural network function approximation, KL-CTD can exhibit drastically different behaviour to earlier studied algorithms such as Cramer CTD and classical TD learning.
>
> We emphasize that KL-CTD is the original CTD algorithm, as described at Eqn (6) and Line 141.
>
>
>
> **How to understand the convergence result in this paper based on KL divergence, given that distributional dynamic programming is only non-expansive if we directly employ the KL divergence to measure the distribution distance**
>
> Our convergence analysis relies on Lyapunov stability results, as opposed to writing our updates as a stochastic approximation of a contractive operator.
>
> More specifically, there are several reasons why a result concerning the unprojected distributional Bellman operator's behaviour as measured by KL divergence is not pertinent to our analysis here:
> We are concerned with convergence of an algorithm that maintains a categorical representation of the return distribution, but the result above is concerned with the unprojected distributional Bellman operator.
> While we analyze an algorithm that is defined via the KL loss, this does not obligate us to make use of contraction results measuring distance in KL. In Bellemare et al. (2023, Chapter 5), this difference is emphasised by distinguishing between metrics used for analysis, and metrics used to define algorithms. In our case, we study an algorithm that uses KL between categorical distributions in its definition, and our analysis makes use of both KL (to measure distance from the fixed point), and Cramér distance (we make use of contractivity of the categorical distributional Bellman operator in Cramér distance to show that the KL is a Lyapunov function; see the proof of Proposition 5.4 and line 783 in particular).
>
> Please let us know if you have any further queries on this point.
>
>
> **$\tilde{p}$ above Equation (8)**
>
> The $\tilde{p}^\pi_i$ appearing in the equation $\tilde{\eta}^\pi = \sum_i \tilde{p}^\pi_i \delta_{z_i}$ represents the probabilities of the fixed point of the projected distributional Bellman operator.
>
> ​​
> **It is not clear why C is introduced in Eq.9 and 10. Could any intuitive explanation be provided?**
>
> Our previous calculation on lines 225-227 shows that the KL divergence from the current estimate to the fixed point may be increasing under the KL-CTD dynamics. But from this expression, in light of Proposition 5.3., we can reverse-engineer a modification to the updates which would guarantee the decrease of the KL, and lead to convergence. This modification is exactly the preconditioning using $C^\top C$.
>
>
> **It seems that the superiority of the proposed algorithms PKL-CTD over KL-CTD and TD highly depends on the step size.**
>
> The reviewer is correct that which algorithm is best between the various considered in Figure 7 depends on the step size, however we would like to emphasize that at least in the settings in the figure, the dependence of PKL-CTD on step size has a generally equal or smoother curve than the dependence of KL-CTD and TD on step size, which both have deep learning equivalents which are highly used in practice (DQN and C51).
>
>
> **Why consider the asymptotic normality/variance in value estimates in Propositions 6.1 and 6.4? What is the main regularity condition? Is there any potential application of these asymptotic properties?**
>
> We study the asymptotic distributions of the value estimates because they are able to give us exact forms of the errors incurred by the algorithms. These asymptotic results have led to the bias/variance results in our paper, and we believe that they have further applicability for deriving insights in future work.
>
>
> **Comparison with Rowland (2018)**
>
>
> While Rowland et al. (2018) study categorical dynamic programming and the Cramer-CTD update, this paper focuses on the categorical TD learning algorithm based on the gradient of the KL loss, matching the form of loss used in the original categorical distributional RL paper (Bellemare et al., 2017).

---

> > ### Comment · Reviewer_wwL4 · 2025-04-02
> >
> > Thank you for the explanations and some of them are helpful. However, I think my major concerns have not fully been addressed. Here are some follow-up questions.
> >
> > * I understand the authors' claim that the KL-CTD is exactly the TD update of categorical distributional RL, is that right? If so, maybe the paper has mentioned, but I am still curious what the exact gap is between the previously proposed cramer CTD and KL-CTD analytically. I understand the behavior difference in the illustrative experiment, which may be less convincing as it is conducted only in a certain environment, but I think this difference may not be highlighted sufficiently in the current paper. Additionally, despite this gap, is it meaningful enough for practitioners to choose to use the revised CTD in a large-scale environment (not just the illustrative environment) instead of the Cramer CTD? It seems that the motivation in this paper is on the theory side.
> >
> > * In the authors' response,  this paper uses a novel analysis tool based on  Lyapunov stability. It is great to do that, but my concern is why not directly on the KL function or what is the challenge if we still insist on the classical stochastic approximation analysis? I agree this does not obligate the authors to make use of contraction results measuring distance in KL, but I am wondering how the novel analysis tool can circumvent the issues when using the classical analysis tool? This is more suggested to be highlighted in the paper.
> >
> > * I also agree with other reviewers' comments: the experimental section can be largely enhanced in the future, although for a theory paper, it is not a critical issue.
> >
> > * Some properties are fine to present, including asymptotic normality, even though the authors also agree that they may be useful in future work. However, I suggest that the authors make the theoretical results of the paper more connected and highlight the main conclusion they want to make. Unfortunately, I think there is still room for improvement in the current paper regarding this point.
> >
> > In summary, I am inclined to keep my rating for now.

---

> > > ### Author Response · Authors · 2025-04-03
> > >
> > > Thank you very much for the additional questions, we provide responses in turn below.
> > >
> > > **The KL-CTD algorithm.**
> > >
> > > - You're correct, the KL-CTD update we study in this paper, defined in Eqn (6), exactly matches the update proposed by Bellemare et al. (2017) (Algorithm 1 of their paper writes the loss as a cross-entropy, the gradient of which is identical to the KL appearing in Eqn (6)).
> > > - In contrast, the Cramer-CTD algorithm, which is analyzed by Rowland et al. (2018), and summarised at the very end of Section 2 in our paper, does not exactly match the algorithm of Bellemare et al. (2017): the probabilities are not parametrised with a softmax, and updates are not computed via gradients of a cross-entropy/KL loss. We hope this makes clear the analytic difference between KL-CTD and Cramer-CTD, and welcome any further questions.
> > > - Having established that Cramer-CTD and KL-CTD are given by distinct update rules/parametrizations in Section 2, Section 3 provides motivating examples of distinct behaviour in practice. Indeed, as the reviewer pointed out these examples are only in a particular environment, we chose to do this to to the analysis of Cramer-CTD by Lyle et al. (2019), which showed that Cramer-CTD is *always* equivalent to TD in tabular settings. Our onus of proof in light of their results was to show that *there exists* settings where the algorithms are different in practice, to motivate our study.
> > > - "Is it meaningful enough for practitioners to choose to use the revised CTD in a large-scale environment". We want to emphasize that KL-CTD is the method that is typically used in large-scale applications, beginning with the work of Bellemare et al. (2017), and this is an important motivation for aiming to obtain a theoretical understanding of this algorithm.
> > >
> > > **On classical stochastic approximation techniques.**
> > >
> > > - If we understand correctly, one possible interpretation of your suggestion would be to attempt to perform an analysis like that of Jaakola et al. (1994) or Tsitsiklis (1994) for Q-learning, where we aim to interpret KL-based CTD updates as approximating the application of a contractive operator, and use the stochastic approximation results described in these papers to derive a convergence guarantee. This is the approach taken by Rowland et al. (2018) for Cramer-CTD.
> > > - However, it is not clear whether the KL-based CTD updates can be written in this way, as writing the right-hand-side as the application of an operator becomes rather complicated, we can write out the argument in more detail if the reviewer would like to see it. Furthermore, we believe that this difficulty is one of the reasons that the convergence of KL-CTD remains an open question 8 years after it was originally introduced in Bellemare et al. (2017).
> > > - We want to emphasize that this is a commonly encountered scenario when analyzing RL algorithms. For example, this is the situation encountered when analysing TD with linear function approximation (Tsitsiklis and Van Roy, 1997), and the proofs of convergence for linear on-policy TD rely on Lyapunov stability analysis (Tsitsiklis and Van Roy rely on this approach in establishing their Theorem 2, making use of Lyapunov stability results from the textbook by Benveniste, Métivier, Prioret (1990)). As another example, in distributional RL, Rowland et al. (2024) made use of Lyapunov stability theory described by Benaïm et al. (2005) to prove convergence of quantile TD.
> > > - We are not sure we fully understand what the proposal to use classical stochastic approximation techniques would look like, would you be able to provide more detail?
> > >
> > > **Experiments.**
> > > - Thank you for this comment: We agree that there are promising directions for future empirical study, particularly of the PKL-CTD algorithm proposed in this paper, and also share the view that as the paper is primarily theoretical, our emphasis has been on establishing theoretical understanding, rather than an empirically-focused work.
> > >
> > > **Organisation**
> > > - We acknowledge that the sequence of results in our paper is likely better represented as a tree rather than a linear chain. However we believe this to be a strength rather than a weakness, as this approach lets us present multiple key insights within a single body of work. This is particularly beneficial for future research building on of our results, as it opens multiple avenues which can be expanded upon. That said, we will revisit the presentation of our results, and see if there’s anywhere we can make the connection between results more explicit, or more clearly guide the reader through our findings.
> > >
> > >
> > > **References**
> > >
> > > A. Benveniste, M. Métivier, and P. Prioret, Adaptive Algorithms and Stochastic Approximations. Berlin: Springer-Verlag, 1990.
> > >
> > > Michel Benaïm, Josef Hofbauer, and Sylvain Sorin. Stochastic approximations and differential inclusions. SIAM Journal on Control and Optimization, 44(1):328–348, 2005.

---

### Official Review · Reviewer_UawD · 2025-03-12

**Overall Recommendation:** 2

**Summary:**

The paper studies categorical distributional reinforcement learning  with a KL divergence loss. Unlike previous analyses relying on the Cramér distance, this paper introduces a novel preconditioned version of categorical temporal-difference learning with KL divergence , proving its convergence under mild assumptions. The paper also analyzes the asymptotic variance behavior of categorical estimates under various learning rate schedules. Empirical evaluations demonstrate that KL-based algorithms perform differently from classical temporal-difference methods in specific tabular reinforcement learning environments.

**Claims And Evidence:**

The main claims inluding convergence of PKL-CTD, the asymptotic variance analysis, and advantages of KL-based methods are clearly supported by theoretical proofs and empirical experiments. However, the manuscript occasionally lacks clarity, especially around certain critical derivations (e.g., derivation of equation (6) from equation (5)). Additionally, the authors should more clearly differentiate their contributions from prior work. I do have some additional concerns
- The paper provides a counterexample showing KL divergence does not serve as a Lyapunov function, but that does not exclude the possibility of convergence of KL-CTD without preconditioning.
- The paper presents empirical results showing distinct learning dynamics for KL-CTD. However, the theoretical justification is somewhat heuristic, and the paper lacks a rigorous explanation for why KL-CTD is preferred beyond empirical observations.
- The clarity regarding the derivation of certain critical equations (such as equation (6)) needs improvement.

**Essential References Not Discussed:**

No

**Experimental Designs Or Analyses:**

The experimental analyses  illustrate the theoretical findings. Experiments comparing TD, KL-CTD, and PKL-CTD are well-designed and show that KL-CTD outperforms TD in high-stochasticity environments, while PKL-CTD is the most stable.

**Methods And Evaluation Criteria:**

The proposed methods (PKL-CTD and KL-CTD) and evaluation criteria, which uses standard RL benchmarks (Cycle, Garnet, Dirichlet environments), are reasonable and appropriate for assessing theoretical claims. However, the analysis is restricted to synchronous updates and policy evaluation settings without exploring more interesting setting like asynchronous or control (Q-learning), which limits the generality of the results.

**Other Comments Or Suggestions:**

- Clarify the derivation from equation (5) to equation (6) explicitly in the main text or clearly reference its derivation in supplementary material.
- Highlight the key differences/novelty from Rowland et al. (2018) and Boeck & Heitzinger (2022).
- Provide more intuition for PKL-CTD by explaining why preconditioning improves convergence.
- It might be insightful to discuss the potential impacts of different choices of the preconditioning matrix on convergence behavior and practical performance.
- Indicates where the proof can be found
- Proposition 4.3. The author should clarify that the stationary point is fixed point of Projectied Bellman operator
- The equation in Line 224-228 is not clear
Some typos
- $|| ||_{\pi}$ is not defined in Proposition 5.3.
- Proposition 6.1. middle line $\tilde{p}$

**Other Strengths And Weaknesses:**

Strengths
- Theoretical advancement by addressing the convergence gap for KL-based distributional RL, which partially solved the problem raised by  Dabney et al., 2018
- Asymptotic variance analysis is solid
- Connects theory with empirical observations, particularly in variance behavior.
- Derives insights on learning rate selection and category scaling

Weakness
-  Analysis is restricted to synchronous policy evaluation scenarios, not covering asynchronous and control setting
- The proof of Theorem 5.5 is largely standard stochastic approximation theory, meaning the novelty lies more in the application rather than the mathematical difficulty.
- Minor clarity issues in crucial derivations.

**Questions For Authors:**

- Have you explored  the convergence and performance of KL-CTD or PKL-CTD with asynchronous updates and control setting?
- While the paper provides a counterexample showing KL fails to be a Lyapunov function for KL-CTD, but that does not exclude the possibility of convergence of KL-CTD without preconditioning.

**Relation To Broader Scientific Literature:**

The paper extends prior work on distributional RL (Bellemare et al., 2017) and categorical RL (Rowland et al., 2018). It is related to Cramér-based CDRL (Boeck & Heitzinger, 2022) but focuses on KL divergence instead. Some recent works on quantile-based DRL are not cited.

**Theoretical Claims:**

The main convergence proofs appear correct and leverage standard stochastic approximation techniques and existing convergence theory. No technical flaws were found, though the techniques used are relatively standard and do not introduce significant novel technical complexities.

---

> ### Author Rebuttal · Authors · 2025-04-01
>
> We thank the reviewer for their time and effort in reviewing our paper, and for the helpful feedback provided. We are pleased that the reviewer found our work on the convergence of PKL-CTD, the asymptotic variance analysis, and the connection between theory and empirical observations to be valuable. Below, we address each of the reviewer’s points in detail.
>
>
> **Analysis is restricted to synchronous policy evaluation scenarios**
>
> *Synchronous/Asynchronous*
>
> We note that all experimental results in Section 7 are done using asynchronous updates (with synchronous updates performed as an ablation in Appendix F), though the reviewer is correct that our convergence result addresses the synchronous setting. In light of the reviewer’s suggestions we have expanded our analysis, and give a sketch below of how it can be applied to analyse asynchronous cases too.
>
> Following work presenting core convergence guarantees in asynchronous stochastic approximation, such as Borkar (2008), considering the case where states are updated according to an ergodic Markov chain with stationary distribution $c$, the key concern to establishing convergence under appropriate technical conditions on step sizes is to establish a corresponding Lyapunov function for the continuous-time dynamics
>
> $$ \partial_t \phi_t (x) = c(x) C^\top C (T^\pi - I) p^{\phi_t}(x),  $$
>
> which take into account the average per-state weighting of updates. From our analysis of the synchronous case (see Proposition 5.4), it can then be verified that $L(\phi) = \sum_{x\in\mathcal{X}} \frac{d^\pi(x)}{c(x)} \mathrm{KL}(\tilde{p}^\pi(x) \| p^\phi(x))$ is a Lyapunov equation for this ODE, which can then be used to guarantee its convergence.
>
> *Policy Evaluation/Control*
>
> We focus on the policy evaluation setting in this paper, and leave the control setting as interesting future work. We remark that understanding the policy evaluation setting first is typical in the analysis of both standard and distributional reinforcement learning.
>
>
> **Proof techniques used are relatively standard, existing stochastic approximation theory**
>
> We agree that our proof of Theorem 5.5 uses existing stochastic approximation theory, although we would argue that many applications of stochastic approximation theory to RL make use of core existing theory (such as in the classic papers analyzing tabular TD algorithms by Jaakkola et al. (1994) and the linear TD analysis of Tsitsiklis and Van Roy (1997)), and indeed novel theory and techniques of stochastic approximation are often published as stochastic approximation results in their own right.
>
> Further, we want to emphasize that our theoretical analysis leads to a number of novel results, such as an undiscovered algorithm (PKL-CTD), the effective learning rate phenomenon which we show is fundamental to KL-based categorical algorithms, and the bias-variance tradeoff in the atom locations.
>
>
> **Convergence of non-preconditioned KL-CTD**
>
> We agree that the reviewer is entirely correct that our counterexample to the KL being a Lyapunov function for non-preconditioned KL-CTD does not exclude its possibility of convergence, and we will make this clearer in the paper. We chose to include this as an additional result in the appendix as it is a natural question in light of Proposition 5.4 (does the weighted KL, our exhibited Lyapunov function for PKL-CTD, also work as a Lyapunov function for KL-CTD?), we did not intend to suggest that non-preconditioned KL-CTD cannot converge.
>
>
> **Recent works on quantile-based DRL are not cited**
>
> We would like to ensure that no relevant work is missing from our discussion of related work in Appendix E. We invite the reviewer to suggest any works in particular they find to be missing.
>
>
> **Other Comments Or Suggestions**
>
> We thank the reviewer for all of the additional catches/suggestions, and we will ensure to fix them in the text.
>
> **References**
>
> Jaakkola, Tommi, Michael Jordan, and Satinder Singh. "Convergence of stochastic iterative dynamic programming algorithms." Neural Computation (1994).
>
> Tsitsiklis, John and Van Roy, Ben "An analysis of temporal-difference learning with function approximation," in IEEE Transactions on Automatic Control (1997)
>
> Borkar, Vivek. Stochastic approximation: a dynamical systems viewpoint. Cambridge University Press, (2008).

---

> > ### Comment · Reviewer_UawD · 2025-04-04
> >
> > Thank the authors for the detailed rebuttal and clarifications. However, several core concerns remain only partially addressed to me
> > - The authors’ sketch connects their synchronous analysis to asynchronous updates using Borkar (2008), but the current rebuttal does not provide a rigorous convergence result or even a concrete theorem statement for the asynchronous case. Given that all experiments are performed with asynchronous updates, the absence of a matching theoretical guarantee limits the strength of the claims. A more formal statement or at least a more complete derivation (even in supplementary material) would better support the claim.
> > - While focusing on policy evaluation is reasonable, the paper would benefit from at least discussing challenges in extending to control, especially given the practical importance of distributional RL for control tasks.
> > - The authors argue that their use of existing stochastic approximation theory is consistent with prior reinforcement learning literature. I agree. However, the technical novelty of the convergence analysis is limited, and this remains a weakness of the theoretical component of the paper. While the application to a novel algorithm (PKL-CTD) is worthwhile, I encourage the authors to better emphasize which parts of the analysis, e.g., insights about the Lyapunov structure or atom bias/variance tradeoffs, are new and important, beyond the convergence guarantee itself.
> > - I appreciate the authors’ clarification that the failure of KL to be a Lyapunov function does not imply divergence. However, the current version of the manuscript (prior to the rebuttal) may give readers the incorrect impression that KL-CTD cannot converge. I strongly recommend making this distinction more prominent in the main text (not only in the appendix), along with a discussion of whether practical convergence of KL-CTD is typically observed empirically. Additionally, if the goal is to motivate preconditioning, a deeper discussion on why KL-CTD might be unstable without it, e.g., via spectral properties or gradient scaling, would strengthen the contribution and distinguish it more clearly from previous works.
> >
> > Although I find the core idea  interesting and potentially impactful, I maintain my recommendation.  If revised to address the concerns above, I believe this paper would be a strong candidate for a future venue or for acceptance in its improved form.

---

> > > ### Author Response · Authors · 2025-04-08
> > >
> > > Thank you very much for these additional questions, we add our responses below.
> > >
> > > **Asynchronous analysis**
> > >
> > > We agree that the exact form of the theoretical guarantee is valuable, and we present it below.
> > >
> > > ---
> > >
> > > Suppose that $(\phi_k)_{k\geq 0}$ is a sequence of logits generated according to asynchronous PKL-CTD updates. That is, for
> > > each $k \geq 0$, we receive a transition $(x_k, R^{x_k}, X^{x_k})$ and update
> > >
> > > $$\phi_{k+1}(x_k)  = \phi_k(x_k) + \alpha_k C^\top C \left(\sum_{i=1}^m p(X^{x_k}) h(R^{x_k} + \gamma z_i)- p^{\phi_k}_i(x_k) \right),$$
> > >
> > > and maintain $\phi_{k+1}(x) = \phi_k(x)$ for  $x \ne x_k$.
> > >
> > > Further suppose that the stepsizes $(\alpha_k)_{k \ge 0}$ satisfy the Robbins-Munro conditions
> > >
> > >  $\sum_{k=0}^\infty \alpha_k = \infty$ and  $\sum_{k=0}^\infty \alpha_k^2 < \infty$, $\alpha_{k+1} \leq \alpha_k$ eventually, $\sup_{k\geq 0}\frac{\alpha_{\lfloor zk \rfloor}}{k} <\infty$ for all $z\in(0,1)$, and  $\left(\sum_{i=0}^{\lfloor zk \rfloor} \alpha_{zk}\right) / \left(\sum_{i=0}^{k} \alpha_{k}\right) \to 1$ as $k\to \infty$ for all $z \in (0,1)$. Letting $\nu(x,k)$ be the indicator of whether the state $x$ is updated at step $k$, we further assume that $\lim\inf_{k\to\infty} \frac{\nu(x,k)}{k} \geq \Delta$ for some constant $\Delta > 0$, and defining $N(k,z)=\min\{ n>k: \sum_{i=k+1}^n \alpha_i > z \}$, the limit  $\lim_{k\to\infty}\frac{\sum_{n=\nu(x, k)}^{\nu(x, N(k, z))} }{\sum_{n=\nu(y, k)}^{\nu(y, N(k, z))}}$ exists almost surely for all $x, y\in \mathcal{X}$.
> > >
> > > Then we have that $\phi_k$ converges in the sense that $p^{\phi_k}(x) \to \tilde{p}^\pi(x)$ for every $x\in \mathcal{X}$ such that $d^\pi(x)>0$ almost surely.
> > >
> > > ---
> > >
> > > This result is based on Theorem 2.5 of Borkar and Meyn (2000), and this is only a single example of an asynchronous convergence result which can be obtained. We want to emphasize that different convergence results can be obtained under different choices of assumptions, but the vital component is the existence of a Lyapunov function, which we provided in our original rebuttal. In the setting of Borkar and Meyn (2000), the existence of the Lyapunov function provides us the fact that the fixed point is an asymptotically stable equilibrium.
> > >
> > >
> > > **Extension to control**
> > >
> > > Thank you for this comment: We agree that highlighting the challenges in extending to control is a good idea, and we will highlight this as a direction for future work in the text. To clarify where the complications arise, it is primarily due to the combination of the nonlinear dynamics of the softmax parameterization, and the nonlinear update due to the argmax over actions. Handling each of these nonlinearities individually is relatively straightforward, however their combination brings forward challenges.
> > >
> > >
> > > **Novelty of convergence result**
> > >
> > > We disagree with the reviewer’s statement that “the technical novelty of the convergence analysis is limited, and this remains a weakness of the theoretical component of the paper.” We expand on our reasoning below.
> > > - Firstly, we want to highlight one of the main motivations of our paper is that KL-based distributional RL losses have been ubiquitous in large-scale deep RL experiments since the introduction of C51 in Bellemare et al. (2017), yet they have had little theoretical analysis (to our knowledge none).
> > > - In light of this, we believe the convergence result in our paper to be a significant technical result: as we discussed in our rebuttal to reviewer wwL4, the analysis of dynamics with KL-based losses aren’t straightforward to analyse as the application of a contractive operator, which led us to perform a novel analysis in finding a Lyapunov function for its convergence.
> > > - We will also better emphasize which parts of the analysis, aside from the convergence result itself, are new and important, namely (i) exact quantities for the asymptotic variance of the KL and Cramér value estimators, (ii) the phenomenon of the effective learning rate being scaled proportional to the number of bins, and (iii) the bias-variance tradeoff present in the choice of atom locations.
> > >
> > >
> > > **Convergence/divergence of KL-CTD**
> > >
> > > - We agree that our current presentation does not explicitly state that the lack of the KL as a Lyapunov function for KL-CTD does not indicate that KL-CTD must diverge on the provided counterexample, and we will make this explicit in the main text. We will also make clear that practical convergence of KL-CTD is generally observed.
> > > - As for how to motivate the form of the preconditioning used, we aimed to do exactly that in lines 222-233: in particular we motivated the preconditioner as a way to change the inner product to a weighted inner product in which the quantity is always negative due to the contractivity of $T^\pi$ in the $C$-weighted $\ell^2$ norm.
> > >
> > > **References**
> > >
> > > V. S. Borkar and Sean P Meyn. The ODE method for convergence of stochastic approximation and reinforcement learning. SIAM Journal on Control and Optimization, 38(2):447—469, 2000.

---

### Decision · Program_Chairs · 2025-05-01

**Decision:**

Accept (poster)

**Comment:**

This paper analyses distributional temporal difference learning with categorical representation using KL divergence as loss (KL-CTD). The choice of KL divergence matches the original C51 paper while previous analysis relies on changing the KL divergence to some other loss to facilitate analysis. Based on this fact, I believe this paper is well motivated. That being said, as pointed by the reviewers, this paper does fall short in terms of experiments and considering only synchronous updates in the analysis. I personally believe the extension to asynchronous updates should be simple, especially with i.i.d. samples. Missing this extension does not constitute a major flaw to me. And I appreciate the authors' initial efforts towards this in the response. Since the contribution is mostly theoretical, I will not hold a very high standard for empirical study to require large scale experiments. Two reviewers are negative but I do not see their complain about technical flaws.

Overall, I think this is a sold work that nicely closes a gap in the literature. I therefore recommend accept. That being said, I do encourage the authors to improve the paper according to the reviews.